# Cytosolic aspartate aminotransferase moonlights as a ribosome-binding modulator of Gcn2 activity during oxidative stress

**Robert A Crawford[1], Mark P Ashe[1], Simon J Hubbard[2], Graham D Pavitt[1]\***

[1]Division of Molecular and Cellular Function, Faculty of Biology Medicine and Health, Manchester Academic Health Science Centre, The University of Manchester, Manchester, United Kingdom; [2]Division of Evolution, Infection and Genomics, Faculty of Biology Medicine and Health, Manchester Academic Health Science Centre, The University of Manchester, Manchester, United Kingdom

**\*For correspondence:**
graham.pavitt@manchester.ac.uk

**Competing interest:** The authors declare that no competing interests exist.

**Abstract** Regulation of translation is a fundamental facet of the cellular response to rapidly changing external conditions. Specific RNA-binding proteins (RBPs) co-ordinate the translational regulation of distinct mRNA cohorts during stress. To identify RBPs with previously under-appreciated roles in translational control, we used polysome profiling and mass spectrometry to identify and quantify proteins associated with translating ribosomes in unstressed yeast cells and during oxidative stress and amino acid starvation, which both induce the integrated stress response (ISR). Over 800 proteins were identified across polysome gradient fractions, including ribosomal proteins, translation factors, and many others without previously described translation-related roles, including numerous metabolic enzymes. We identified variations in patterns of PE in both unstressed and stressed cells and identified proteins enriched in heavy polysomes during stress. Genetic screening of polysome-enriched RBPs identified the cytosolic aspartate aminotransferase, Aat2, as a ribosome-associated protein whose deletion conferred growth sensitivity to oxidative stress. Loss of Aat2 caused aberrantly high activation of the ISR via enhanced eIF2α phosphorylation and *GCN4* activation. Importantly, non-catalytic *AAT2* mutants retained polysome association and did not show heightened stress sensitivity. Aat2 therefore has a separate ribosome-associated translational regulatory or 'moonlighting' function that modulates the ISR independent of its aspartate aminotransferase activity.

## Editor's evaluation

In this article, Crawford et al. monitored stress-induced alteration in proteins associated with translating ribosomes in yeast using mass spectrometry-based approaches. This revealed that cytosolic aspartate aminotransferase 2 (Aat2) is associated with polysomes. Aat2 deletion sensitizes yeast to oxidative stress which is paralleled by aberrantly elevated integrated stress response. The authors also show that polysome-association of Aat2 and its role in oxidative stress response are independent of its aminotransferase activity. Altogether, it was found that this study is of broad interest since it provides further evidence that metabolic enzymes may "moonlight" as post-transcriptional regulators while highlighting previously unappreciated aspects of adaptation to stress.

## Introduction

Translation and its regulation are highly complex processes requiring the concerted action of numerous factors associated with mRNAs: translation factors (TFs), tRNAs, ribosomes, and RNA-binding proteins (RBPs) (*Dever et al., 2016*). Importantly, the translation apparatus is nimble in that its regulation immediately impacts upon the protein content of cells to meet changing conditions. Both general and mRNA-specific translational regulatory mechanisms co-ordinate cellular responses to diverse cues (*Jackson et al., 2010*). One translational control pathway common to all eukaryotes is called the integrated stress response (ISR) (*Pakos-Zebrucka et al., 2016*) also known as general amino acid control in yeast. Distinct cellular stress signals activate one or more of a family of protein kinases (GCN2, PKR, PERK, and HRI) that each phosphorylate the general translation initiation factor eIF2 on a conserved single serine residue of its alpha subunit (*Wek, 2018*). This causes a rapid global downregulation of protein synthesis. Under active translation conditions, eIF2 in its GTP-bound form recruits initiator tRNA to ribosomes. Phosphorylated eIF2 instead forms an inhibited complex with the key guanine nucleotide exchange factor eIF2B, which otherwise generates active eIF2-GTP (*Pavitt, 2018*; *Adomavicius et al., 2019*; *Kashiwagi et al., 2019*). However, not all translation is inhibited by the ISR. mRNAs encoding stress-protective proteins need to be translated for cells to adapt to the altered cellular environment. The 5′ untranslated regions (5′UTRs) of some translationally controlled mRNAs contain *cis*-acting elements promoting their translation when bulk protein synthesis is attenuated (*Hinnebusch et al., 2016*; *Wek, 2018*). Key examples include the mammalian *ATF4* and yeast *GCN4* mRNAs, which are both transcriptional activators whose expression is controlled via regulated ribosome reinitiation at upstream open reading frames (uORFs) (*Hinnebusch et al., 2016*). Once the stress has been neutralised, translation patterns revert to steady state, restoring proteostasis (*Crawford and Pavitt, 2019*). In the ISR, this requires the dephosphorylation of eIF2 (*Wek, 2018*). In humans a failure to restore proteostasis in a timely manner can contribute to a range of diseases, including cognitive disorders and cancer (*Costa-Mattioli and Walter, 2020*).

Other stress-dependent mechanisms also operate on translation. Inhibiting RNA helicases that promote ribosome recruitment and the unwinding of secondary structures during 5′UTR scanning has been shown to reduce translation significantly (*Sen et al., 2015*). In yeast, both glucose starvation and heat shock stresses are accompanied by a dramatic reduction in mRNA binding by both the eIF4A and Ded1 RNA helicases (*Castelli et al., 2011*; *Bresson et al., 2020*), which contributes to very rapid translational repression, within 1 min following glucose withdrawal (*Ashe et al., 2000*). In addition, translation elongation can be regulated through modulation of the activity of elongation factors such as eEF2, which is modified downstream of some stress signalling pathways, including oxidative stress. tRNA availability also controls elongation rates. Local variations in codon usage and ribosomal pausing and stalling events all slow elongation (*Dever and Green, 2012*; *Schuller et al., 2017*; *Wu et al., 2019*; *Tesina et al., 2020*). Stalled ribosomes lead to ribosome collisions, which can activate Gcn2 and the ISR (*Wu et al., 2020*; *Pochopien et al., 2021*; *Yan and Zaher, 2021*) as well as ribosome-associated quality control (RQC) pathways that recycle stalled ribosomes and degrade defective mRNAs (*D'Orazio and Green, 2021*).

Ribosomes themselves are not necessarily uniform and can vary between conditions and cell types (*Slavov et al., 2015*). Variation in ribosome composition can confer preferences for binding to different subsets of mRNAs. Examples include the ribosomal protein (RP) Rps26/eS26, where both high salt and raised pH reduce its incorporation into yeast ribosomes. Rps26-deficient ribosomes are proposed to preferentially translate stress-responsive mRNAs via altered Kozak sequence recognition preferences (*Ferretti et al., 2017*). Furthermore, some paralogous RPs, where two genes encode the same RP, have been demonstrated to have specific roles in translational regulation. For example, yeast Rpl1a/uL1 and Rpl1b show preferences for translating different sets of mRNAs, as Rpl1b-containing ribosomes promote more efficient translation of mitochondrial proteins required for respiratory growth (*Segev and Gerst, 2018*). Mutations in RPs lead to Diamond–Blackfan Anemia and other ribosomopathies in humans, which might imply specialised roles for RP-deficient or paralog-specific ribosomes. Alternatively, reducing the abundance of active ribosomes via defects in RPs likely changes the balance of expression of mRNAs observed in different cells or tissues, potentially contributing to ribosomopathies (*Mills and Green, 2017*).

RBPs and ribosome-interacting proteins also contribute to translational control. Several RBPs have been observed to modulate the expression of sets of target mRNAs, which range from a few target

transcripts to several thousand (*Hogan et al., 2008*). For example, the RBP CPEB is only recruited to mRNAs containing a cytoplasmic polyadenylation element and helps to modulate ribosome recruitment and translation (*Richter, 2007*). Multiple studies have attempted to either identify mRNA targets of specific RBPs (*Hogan et al., 2008*) or identify new RBPs across a range of organisms, using a variety of methods (*Hentze et al., 2018*). Curiously, these latter studies have uncovered an unexpectedly large number of metabolic enzymes that bind RNA. Very few of these newly recognised RBPs have a defined role in RNA biology, but it has been suggested that many could have a second 'moonlighting' function when bound to RNA. These results highlight the possibility that undiscovered post-transcriptional regulatory networks link gene expression and intermediary metabolism (*Hentze and Preiss, 2010*).

A model has therefore emerged of a complex interplay between mRNA-specific elements, RBPs, and ribosome-associated factors that may combine to enable selected stress-responsive mRNAs to escape globally repressive regulatory mechanisms. Here, we set out to identify candidate proteins that might be involved in mRNA-specific translational regulation during stress in yeast. Oxidative stress caused by the addition of hydrogen peroxide ($H_2O_2$) brings about rapid translational repression via Gcn2 activation and the ISR (*Shenton et al., 2006*), as well as the induction of antioxidant enzymes at both the transcriptional and translational levels to ameliorate the stress (*Morano et al., 2012*). How these antioxidant enzymes are translated under such repressive conditions is unclear. We took an unbiased proteomics approach to determine patterns of ribosome association across polysome gradients in actively growing cells and how these change in response to $H_2O_2$. In parallel, we assessed how cells respond to amino acid starvation caused by the addition of 3-amino-1,2,4-triazole (3-AT), a well-characterised inhibitor of histidine biosynthesis (*Hinnebusch, 2005*). Both stresses activate the yeast ISR and translation of *GCN4* (Hinnebusch; *Shenton et al., 2006*) but also need to promote the expression of different stress-specific genes. We find that the changes in polysome enrichment (PE) of proteins in response to both stresses are remarkably well correlated, with similar changes observed across both stresses for TFs, RPs, and a wide range of RBPs including metabolic enzymes. While some RBPs closely followed the PE profiles of RPs during stress by accumulating in the 80S/monosomal fraction, other proteins maintained or enhanced their association with the remaining heavy polysomes during the stress response. By screening candidate RBP knockout strains for oxidative stress phenotypes, we identified cytoplasmic aspartate amino transferase (Aat2) as a novel ribosome-interacting protein with a 'moonlighting' function. We show that Aat2 binds to 60S ribosomes and moderates Gcn2 activation in response to $H_2O_2$. Remarkably, mutational analyses show that the stress-response role is independent of its metabolic role.

## Results

### Quantification of co-ordinated alterations in polysome association of proteins during acute stress

To gain insight into mechanisms of stress responses, we set out to identify candidate proteins that might be involved in mRNA-specific translational regulation during stress as cells adapt to their changing environment. Proteins involved in translational regulation have been found associated with the translational machinery by previous mass spectrometry (MS) approaches (*Fleischer et al., 2006*), and by targeted studies of individual proteins which have high PE in stressed conditions (*Li et al., 2004*; *Hirschmann et al., 2014*; *Kershaw et al., 2015*). Proteins with mRNA-specific activation roles may retain or enhance their PE during stress. Similarly, RQC factors should be ribosome associated during stress, as both $H_2O_2$ and 3-AT have been associated with enhanced slowing and stalling of ribosomes which recruits the RQC machinery and activates Gcn2 (*Shenton et al., 2006*; *Meydan and Guydosh, 2020*; *Yan and Zaher, 2021*). In contrast, the rapid loss of factors from polysomes may contribute to translational repression, as observed for eIF4A during glucose starvation and heat shock (*Castelli et al., 2011*; *Bresson et al., 2020*).

We initially compared the growth and polysome profile responses of a histidine prototrophic version of the standard yeast lab strain, BY4741, under two stress conditions: acute oxidative stress induced by $H_2O_2$ and amino acid starvation induced by addition of the His3 inhibitor 3-AT. Each stressor was added during exponential growth (*Figure 1A*), causing an interruption to cell growth which caused a loss of polysomes by 15 min (*Figure 1B, C*), in accord with prior studies (*Costello*

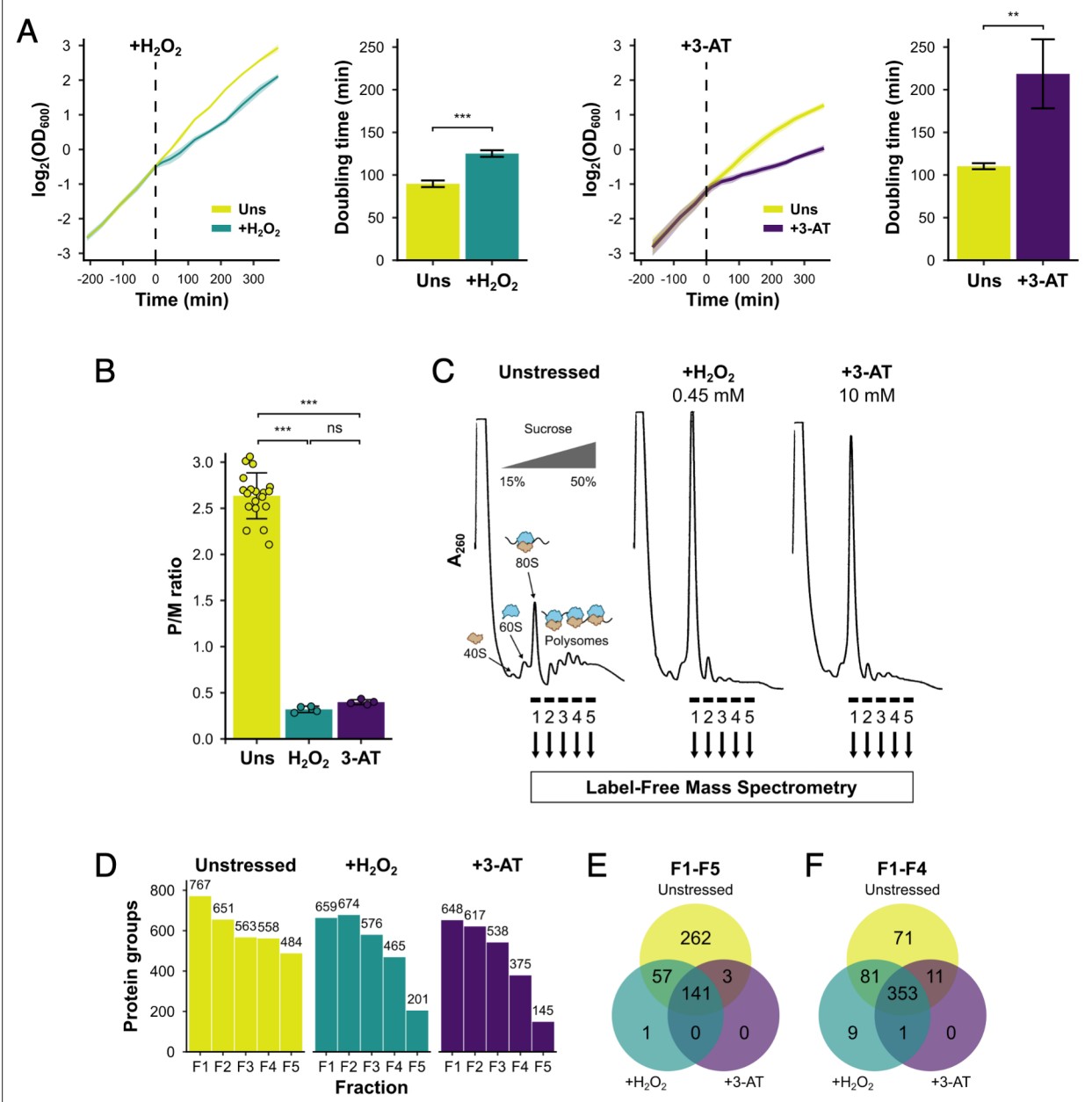

**Figure 1.** Identification of translational regulators using polysomal proteomics. (**A**) Growth curves and doubling times for unstressed, 0.45 mM hydrogen peroxide ($H_2O_2$)-treated and 10 mM 3-amino-1,2,4-triazole (3-AT)-treated cultures ($n = 3$). The time of $H_2O_2$ or 3-AT addition is indicated. (**B**) Quantification of polysome-to-monosome (P/M) ratios under the three conditions ($n = 4$–19). Error bars show standard deviation (SD). The *t*-test was used to compare the conditions: ns – not significant ($p > 0.05$), **$p < 0.01$, ***$p < 0.001$. (**C**) Overview of polysomal proteomics. Monosomal (F1) and polysomal (F2–F5) fractions were isolated from unstressed, $H_2O_2$- and 3-AT-treated extracts and analysed using label-free mass spectrometry (MS). (**D**) The number of proteins identified reproducibly (≥2 replicates) in each fraction. Venn-style diagrams of overlaps between conditions for proteins found across (**E**) all five fractions or (**F**) the first four fractions.

The online version of this article includes the following figure supplement(s) for figure 1:

**Figure supplement 1.** Formaldehyde crosslinking prevents ribosome run-off and retains RNA-binding proteins (RBPs) in polysome fractions similar to cycloheximide treatment.

**Figure supplement 2.** Polysomal proteomics data are reproducible and can separate the different fractions.

**Figure supplement 3.** Nascent peptides are not a major contributor to protein signal.

*et al., 2017*). Growth recovered with time, showing that the cells were able to adapt to each stressor (*Figure 1A*). Qualitative analysis of protein migration through 15–50% sucrose gradients by sodium dodecyl sulphate–polyacrylamide gel electrophoresis (SDS–PAGE) indicated that most proteins were restricted to the top of the gradient, but that specific proteins did migrate deep into the gradient and were retained on polysomes (*Figure 1—figure supplement 1A*).

We employed polysomal proteomics to identify changes in PE for ribosome-associated proteins following treatment with either $H_2O_2$ or 3-AT. Proteins in sucrose gradient fractions were identified and quantified using label-free MS (*Aebersold and Mann, 2016*), from which the PE of each in unstressed and stressed conditions could be determined. To identify such proteins we grew cell cultures to logarithmic phase ($OD_{600}$=0.6) and added 0.45 mM $H_2O_2$ or 10 mM 3-AT for 15 min, as these treatments had equivalent impact on ribosome run-off in polysome profiles (*Figure 1B, C*). We used formaldehyde treatment to stabilise ribosome-associated proteins in polysomes, which gave similar results to cycloheximide fixation for known polysome-associated proteins we assessed by immunoblotting (eIF4E, Puf3, Rps3/uS3, and Rpl35/uL29; *Figure 1*, *Figure 1—figure supplement 1B, C*). Each sample was fractionated on sucrose gradients and five fractions (F1–F5) were analysed using label-free MS (*Aebersold and Mann, 2016*), where F1 represents 80S/monosomal complexes and F2–F5 contain polysomal complexes of increasing size, from disomes to heavy polysomes (*Figure 1B*). For each fraction, as well as the respective unfractionated cytoplasmic lysates (Totals, T), four biological replicates were analysed (originating from separate yeast colonies). Between 145 and 767 unique proteins were identified in at least two replicates of each gradient fraction (*Figure 1D*, *Supplementary file 1* – sheet 2). There was excellent correlation of the measured protein signal between replicates for unstressed fractions F1–F5 ($r^2$ typically >0.9), but more divergence for stressed samples, especially in F4–F5 where, as expected from global polysome profiles, fewer proteins were found migrating deep into the sucrose gradients (*Figure 1—figure supplement 2A*). Principal component analysis showed good clustering of replicates for all three conditions, with adjacent sucrose gradient fractions indicating a gradual change in their composition when moving in sequential order from monosomes (F1) to heavy polysomes (F5; *Figure 1—figure supplement 2B*). Overall, these comparisons indicated that the experimental method was reproducible.

In unstressed conditions, 463 proteins were detected reproducibly in every fraction F1–F5, which rose to 516 when F5 was excluded (i.e. the same 516 proteins were identified in each fraction F1–F4; *Figure 1E, F*). Fewer proteins were found in heavy polysomes during stress when there is ribosome run-off. Nevertheless, 353 proteins were identified in common in every fraction F1–F4 across all three conditions (*Figure 1E, F*). The proteins found across F1–F4 included 70 RPs, 25 TFs and regulatory proteins (TFs), 240 other known RBPs, as recently defined (*Hentze et al., 2018*), as well as 18 proteins that were not previously designated as RNA- or ribosome-binding (*Figure 2A*, *Supplementary file 1* – sheet 2).

A comparison of protein abundance in cytoplasmic extracts showed minimal change in label-free quantification (LFQ) intensity (*Tyanova et al., 2016*) between stressed and control samples, indicating that the dramatic ribosome run-off within 15 min following stress did not have sufficient time to impact the total cellular protein content and that our method sampled cells during their adaptive phase (*Figure 1—figure supplement 2C*). We also saw no evidence that either nascent protein chains or co-translational complex formation (*Shiber et al., 2018*) contributed significantly to the signal for our polysome-associated proteins. Cumulative amino- to carboxy-terminal peptide intensity profiles for individual proteins in total and ribosomal fractions showed no N-terminal bias in F1–F5 compared with T samples, except for rare examples (*Figure 1—figure supplement 3*, see Methods and figure legend for details).

For each protein, we estimated its percentage ribosome association by comparing abundance in the totals and the sum of the ribosome-associated fractions (see Methods; *Figure 2A*, *Supplementary file 1* – sheet 3). As expected, RPs and ribosome-associated chaperones (e.g. Ssz1) were highly associated with ribosomes. In contrast, other proteins showed broad variations in ribosome association, from <2.5% to 100% (*Figure 2A* and *Figure 2—figure supplement 1A*). For abundant and transiently associating translation elongation factors, we estimate less than 20% of each protein was present in F1–F5, while all initiation factors except for the highly abundant eIF4A were over 30% ribosome associated in unstressed cells. Known RBPs (yellow) and proteins not assigned to any other class (grey) showed highly varied ribosome association (*Figure 2A*), reflecting the large functional diversity within

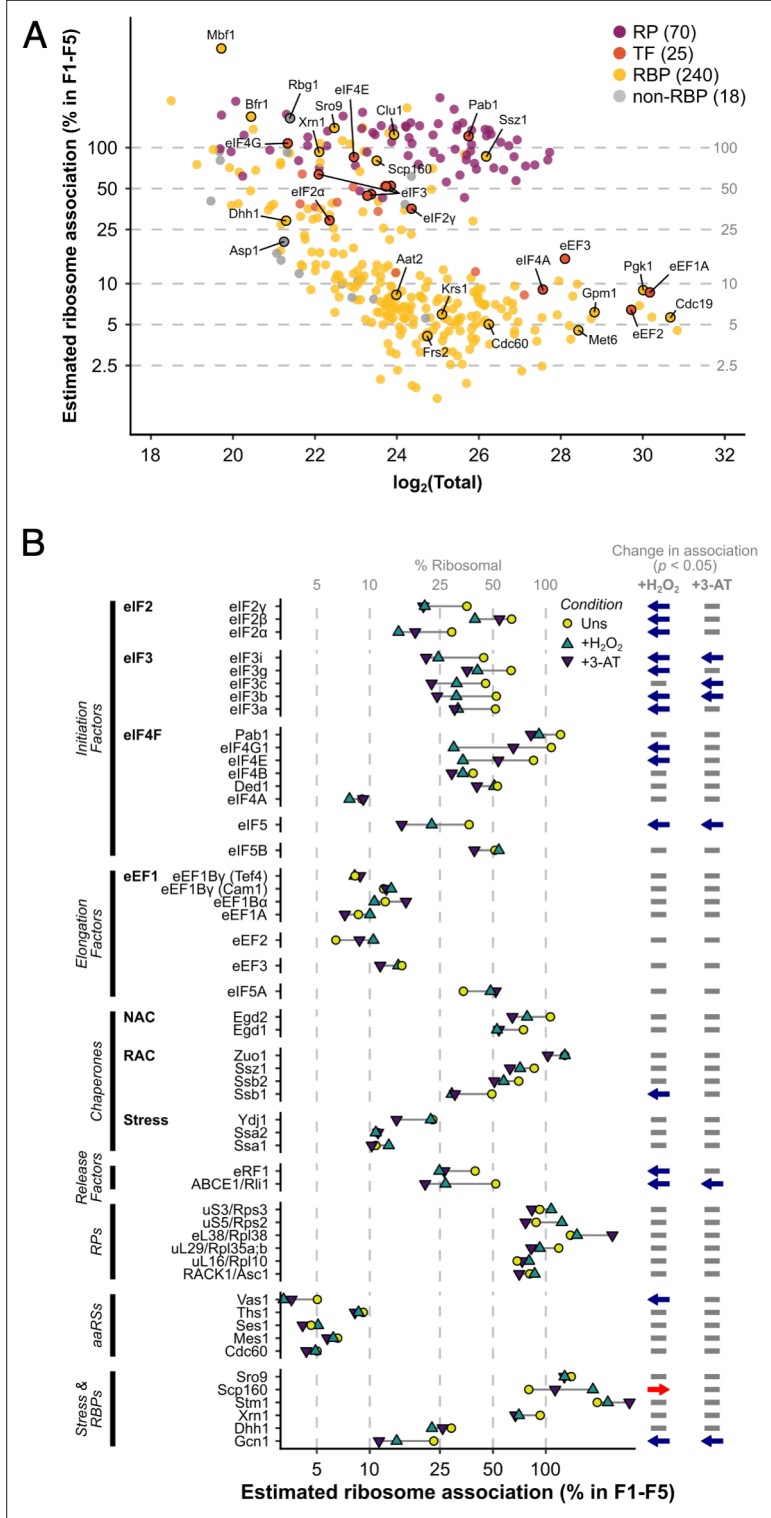

**Figure 2.** The ribosome association of initiation factors decreases during stress. (**A**) Ribosome association of proteins in F1–F4 under all three conditions (353 proteins). The proportion of each protein that is ribosome associated (% ribosomal) was estimated by comparing the summed intensity in the fractions with the totals (see Methods). Protein groups are coloured by their functional category: RP – ribosomal protein, TF – translation factor, RBP – RNA-binding protein, non-RBP – other protein. The number of proteins in each functional category is indicated. (**B**) The change in overall ribosome association during both stresses for selected translation-related

*Figure 2 continued on next page*

*Figure 2 continued*

proteins. Statistically significant changes in ribosome association (p < 0.05) are indicated by arrows: blue – decrease, red – increase.

The online version of this article includes the following figure supplement(s) for figure 2:

**Figure supplement 1.** The ribosome association of initiation factors decreases during stress.

these categories. In response to both stresses, most TFs and related factors either reduced or maintained their overall ribosome association (*Figure 2B* and *Figure 2—figure supplement 1B*). As there is a rapid attenuation in global translation initiation rates, as evidenced by polysome run-off, reduced engagement of initiation factors is expected. Under conditions of glucose starvation or heat shock the RNA helicase eIF4A is rapidly dissociated from polysomes (*Castelli et al., 2011*; *Bresson et al., 2020*). In contrast, we observed no change in eIF4A or Ded1 association, indicating that loss of eIF4A is not a universal response to stress. In addition, a reduction of elongation rates accompanies oxidative stress (*Shenton et al., 2006*), with enhanced eEF2 phosphorylation (*Wu et al., 2020*). Here, we find slightly enhanced association of both eEF2 and eIF5A with ribosomes during both oxidative stress and amino acid starvation, although neither change was statistically significant (*Figure 2B*). These findings are consistent with slowed translocation and enhanced ribosome pausing during stress. In summary, these stress-induced changes in the ribosome association of TFs are consistent with current ideas about how stress impacts the core translation machinery.

## RPs typically respond uniformly to stress

We were able to quantify 55 RPs across F1–F5 in stressed and unstressed cells (*Supplementary file 1* – sheet 2). In unstressed cells, RPs were typically distributed evenly across the monosomal and polysomal fractions (F1–F5), but greater variation was observed during stress, especially in the fractions representing denser polysomes (*Figure 3A*). For example, Rpl38/eL38 was among the most polysome-enriched RPs during stress, with relatively high levels in F5, while under the same conditions Asc1/RACK1 stayed relatively evenly engaged and Rps2/uS5 was significantly depleted from F4 to F5 relative to other RPs (*Figure 3B*).

We used unbiased hierarchical clustering of normalised LFQ intensities to determine 'polysome enrichment profiles' among the 353 protein groups in order to identify enrichment patterns in common across F1–F4 in all conditions (*Figure 1F*). This separated proteins into 14 clusters with variable F1–F4 and/or T enrichments in the presence and absence of stress (*Figure 3C*, *Supplementary file 1* – sheet 3). Proteins in clusters 1–3 exhibited low abundance in ribosome fractions relative to totals, but maintained PE during both stresses. In contrast, those in clusters 11–14 were typically highly associated with ribosomes and enriched in heavier polysome fractions in unstressed conditions, but shifted into the monosome fraction following stress, in line with the global polysome profile change (*Figure 3C*).

Almost all RPs were found in clusters 11 and 12 and generally behaved similar to Rps3/uS3, for example Asc1/RACK1 and Rpl38/eL38 (both cluster 12; *Figure 3B*), consistent with the idea that the RPs are responding in a concerted manner. Within these two clusters, there was some variation in the degree to which the different RPs shifted into monosomes during stress, for example Rps2/uS5 was relatively more depleted from heavy polysomes compared with Rps3/uS3 (*Figure 3B*). Only three RPs separated into other clusters. Rpl8b/uL8 and Rps10/eS10 (cluster 13) appear to be more depleted from heavy polysomes during stress, suggesting that they might be selectively lost from the mRNA-engaged pool of ribosomes during stress (*Figure 3C*). Conversely, Rpl7b/uL30 (cluster 6) retained PE during both stresses in a similar manner to mRNA-binding factors such as Pab1 (*Figure 3C*). Although a growing body of evidence suggests that variations in the association of specific RPs and paralogs with ribosomes contributes to mRNA-specific translation e.g. (*Slavov et al., 2015*; *Ferretti et al., 2017*; *Gerst, 2018*; *Cheng et al., 2019*), our data did not provide strong evidence that variations in RP abundance were critical for these stress responses, so we did not explore the variation we observed among the RPs further. Instead we focused on accessory ribosome-associated proteins.

## RBPs exhibit varied polysome association in response to stress

Several recent genome-wide studies have defined yeast RBPs by crosslinking proteins to RNA and then identifying them by an MS approach or by an in vitro protein array (*Mitchell et al., 2013*; *Beckmann*

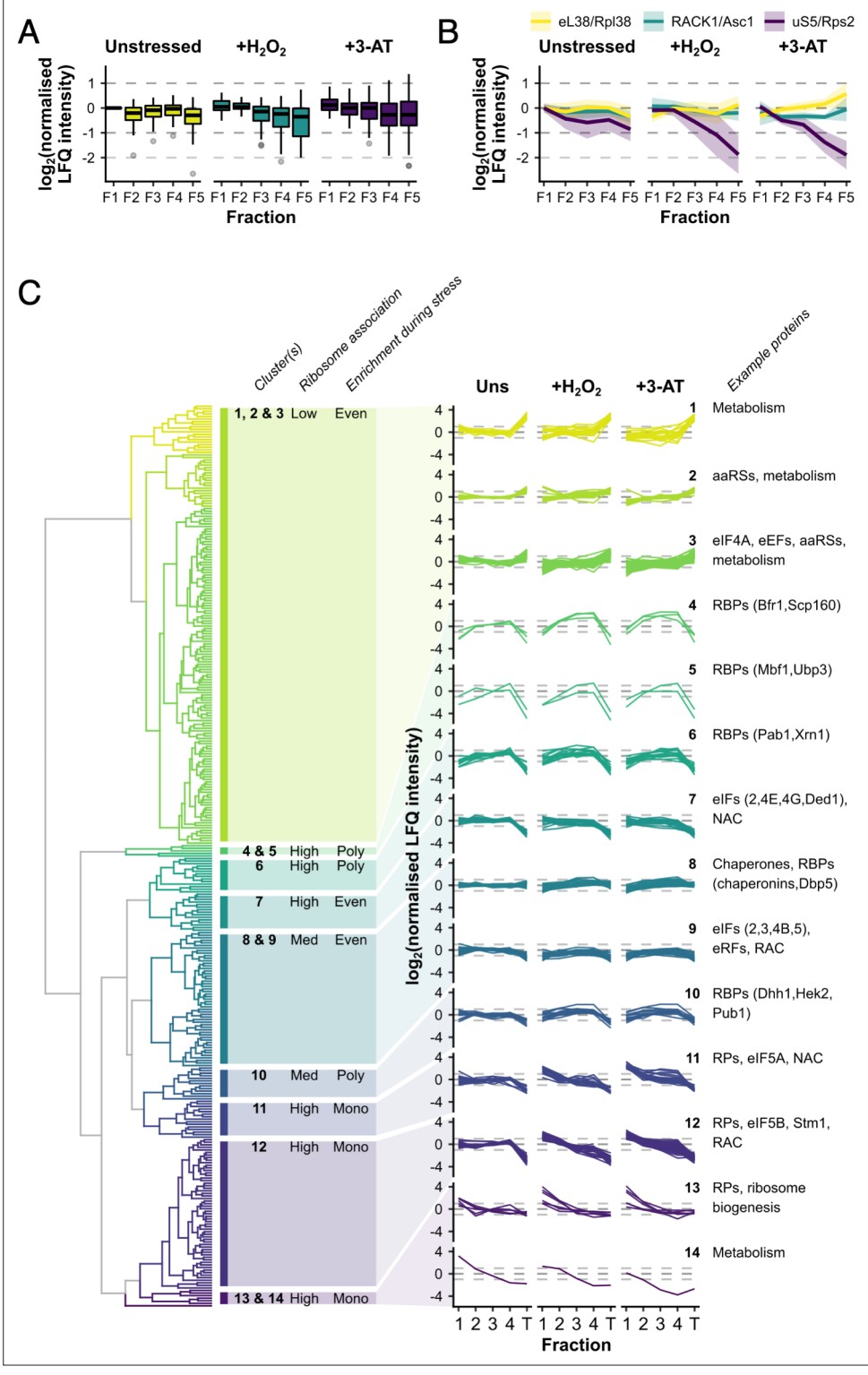

**Figure 3.** Ribosomal proteins (RPs) and RNA-binding proteins (RBPs) show distinct patterns of polysome enrichment. (**A**) The label-free quantification (LFQ) intensities for each RP identified in all conditions F1–F5 were normalised to Rps3/uS3 on a fraction-by-fraction basis and then to unstressed F1. Boxplots showing the distribution of normalised LFQ intensities in each sample. (**B**) Polysome association profiles for three example RPs normalised to Rps3/uS3. Shaded areas show mean ± standard deviation (SD). (**C**) Clustered 'polysome enrichment profiles' for proteins identified under all three conditions in F1–F4. The LFQ intensities for each protein were normalised only to its own mean across the unstressed fractions. T – total. The overall ribosome association and

*Figure 3 continued on next page*

*Figure 3 continued*

PE during stress for each cluster or group of clusters is indicated: Poly – polysome-enriched, Mono – monosome-enriched, Even – equally enriched in all fractions.

The online version of this article includes the following figure supplement(s) for figure 3:

**Figure supplement 1.** Metabolic functions are enriched among RNA-binding proteins (RBPs) identified in polysomal fractions.

*et al., 2015*; *Matia-González et al., 2015*). Such studies have found proteins with typical RNA-binding domains (e.g. RRM, KH, and PUF) as well as a range of proteins lacking classical domains, including numerous metabolic enzymes. A recent review concluded over 1200 yeast proteins had been identified as RBPs in high-throughput studies, with many found by multiple independent methods (*Hentze et al., 2018*). Here, we identified over 240 polysome-associated RBPs in addition to the core RPs and TFs (*Supplementary file 1* – sheets 2 and 3). In contrast to the RPs, which form a relatively tight co-regulated group, RBPs are distributed across all 14 clusters, indicating they differ widely both in their percentage ribosome association (low in clusters 1–3, medium in 8–10 and high in the other clusters) and in how their PE changes in response to stress (*Figure 3C*). In contrast to the RPs, which generally became less abundant in heavier fractions during stress, proteins in clusters 1–3 and 8–9 were evenly spread across the fractions in both unstressed and stressed cells, while factors in clusters 4–6 and 10 became more enriched in the heavier polysome fractions during both stresses (*Figure 3C*, *Supplementary file 1* – sheet 3).

We reasoned that the factors remaining associated with polysomes during stress would likely include those acting to resolve the impact of stress at the translational level. For example, the multi-KH domain-containing protein Scp160 (cluster 4) is known to interact with polysomal mRNAs during both optimal growth conditions and during glucose starvation (*Arribere et al., 2011*). Scp160 behaved equivalently here following both $H_2O_2$ and 3-AT treatment (*Figure 3C*). A recent study proposed that Scp160 enhances the polysome association of codon-optimised transcripts by channelling or recycling tRNAs at ribosomes using successive synonymous codons (*Hirschmann et al., 2014*). Interestingly, aminoacyl tRNA synthetases (aaRSs) were enriched in clusters 2 and 3, which maintain PE during stress (*Figure 3C*).

Several RQC factors were identified in our dataset. For example, Mbf1 (cluster 3) is recruited to stalled disomes where it can prevent frameshifting of collided ribosomes (*Wang et al., 2018*). Similarly, Cdc48 (also cluster 3) is recruited to ribosomes to promote ubiquitination of stalled proteins (*Defenouillère et al., 2013*). In contrast Gcn1, which was recently shown to bind across both stalled and collided ribosome disome partners (*Pochopien et al., 2021*) and is necessary for activation of Gcn2 for eIF2 phosphorylation (*Marton et al., 1997*), is placed in cluster 9, which is evenly associated across polysome fractions under all conditions sampled (*Figure 3C*). In contrast, the ribosome-associated chaperone (RAC) complex members Zuo1 and Ssz1, follow the RP pattern in cluster 12. Thus various proteins associated with resolving stalled/collided ribosomes have distinct patterns of PE following these stresses, likely reflecting their individual roles in RQC.

## Cytosolic aspartate aminotransferase moderates the ISR

Gene ontology analysis revealed that many of the most highly polysome-enriched proteins were metabolic enzymes, including glycolytic enzymes (e.g. Pgk1, Fba1, and Eno1) and amino acid biosynthetic enzymes (e.g. Met6, His4, and Trp5) (*Figure 3C*, *Figure 3—figure supplement 1*, *Supplementary file 1* – sheet 5). These proteins are polysome associated under unstressed conditions and retain or increase both their ribosome association and PE during stress (clusters 1–6 and 8–10). While their association with mRNA is known (*Supplementary file 1* – sheet 3), their additional roles (if any) remain unclear. To assess whether any of these RBPs function in the stress response, we screened a selection of single gene deletion strains for growth phenotypes in the presence of $H_2O_2$. We found that deletion of *AAT2*, the cytosolic aspartate amino transferase, conferred an $H_2O_2$-hypersensitive growth phenotype (*Figure 4A*, *Figure 4—figure supplement 1A*). Aat2 was placed in cluster 3 in our polysomal proteomics, indicating that it is associated evenly throughout the polysome fractions in both unstressed and stressed cells, similar to the profiles for eIF4A and translation elongation factors (*Figure 3C*). Polysome profile analysis revealed that translation was similar in untreated *aat2Δ* and wild-type (parent) cells, but ribosome run-off caused by addition of $H_2O_2$ was more acute in *aat2Δ* at

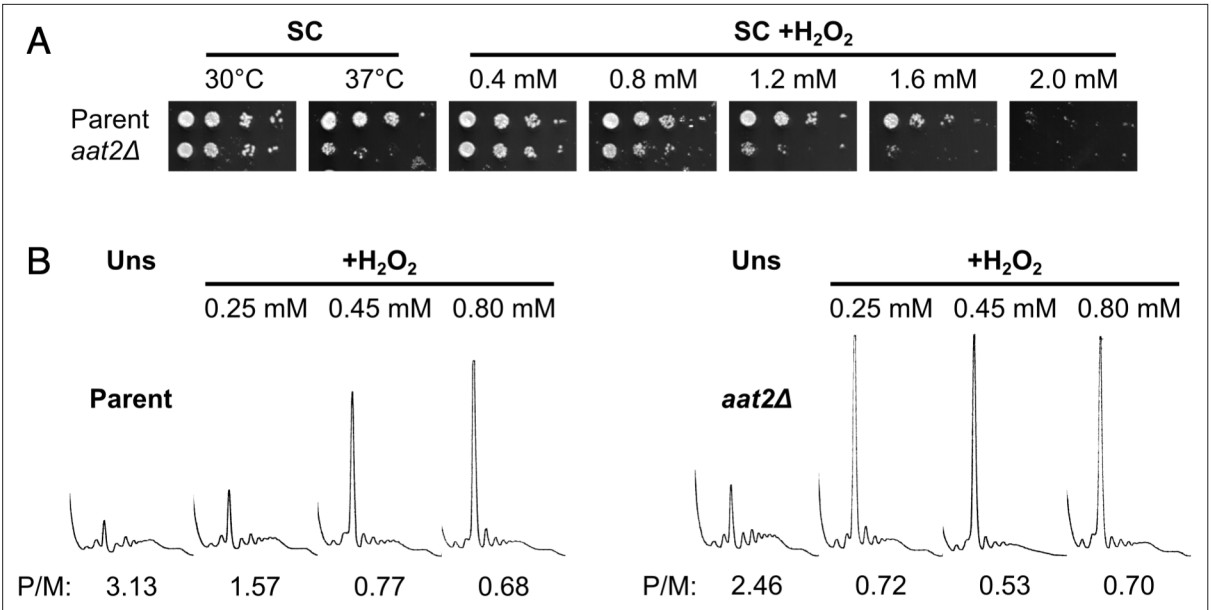

**Figure 4.** Deletion of *AAT2* increases oxidative stress sensitivity. (**A**) Spotting assay of BY4741 parent and *aat2Δ* strains on synthetic complete dextrose (SC) medium in the presence of varying concentrations of hydrogen peroxide ($H_2O_2$). Each spot is a tenfold dilution of the previous one. (**B**) Representative polysome profiles from unstressed and $H_2O_2$-treated cultures. P/M: mean polysome-to-monosome ratio ($n$ = 2–3).

The online version of this article includes the following figure supplement(s) for figure 4:

**Figure supplement 1.** Metabolic functions are enriched among RNA-binding proteins (RBPs) identified in polysomal fractions.

lower concentrations of peroxide (*Figure 4b*, *Figure 4—figure supplement 1B*). This is consistent with reduced translation initiation in *aat2Δ* contributing to $H_2O_2$ growth sensitivity, suggesting that Aat2 has a critical role in translational regulation in response to stress rather than in maintaining steady-state translation. As phosphorylation of eIF2α by Gcn2 is responsible for ribosome run-off following oxidative stress (*Shenton et al., 2006*), we used phospho-specific antibodies to assess the state of eIF2α in whole cell extracts, which showed heightened phosphorylation in response to lower concentrations of $H_2O_2$ in *aat2Δ* cells (*Figure 5A*). The ISR in yeast is mediated via translational control of the transcription factor *GCN4*. The four uORFs in the long *GCN4* 5′ leader limit translation of the main ORF, except where elevated eIF2α phosphorylation inhibits its nucleotide exchange factor eIF2B, enabling uORF skipping and higher Gcn4 expression. We used the well-established *LacZ* reporter plasmid (p180) to indirectly monitor Gcn4 levels in these cells (*Hinnebusch, 2005*; *Gunišová et al., 2018*), which confirmed that *aat2Δ* cells had higher LacZ levels following 2 hr of $H_2O_2$ stress (*Figure 5B*). Control reporter plasmids bearing single or no uORFs were not changed from the parent strain (*Figure 5B*), suggesting there is no defect in mRNA levels, scanning or re-initiation proficiency on the *GCN4* leader. The data indicate that the yeast ISR is aberrantly activated by $H_2O_2$ when Aat2 is deleted.

## Aat2 binds 60S ribosomes

Our polysomal proteomics estimated that 8–10% of Aat2 was ribosome associated (*Figure 2A*, *Supplementary file 1* – sheet 3). A strain bearing a C-terminal TAP tag behaved similarly. In unstressed cell extracts the majority of the tagged protein was retained at the top of a sucrose gradient, with 10% present in fractions 5–9 which correspond to the sucrose gradient fractions analysed by MS (*Figure 6A*). Equivalent percentage ribosome associations were also observed in $H_2O_2$-treated cell extracts using both western blotting with the Aat2-TAP strain and MS with the untagged strain (13% and 9%, respectively; *Figure 6A*, *Supplementary file 1* – sheet 3). Thus, a fraction of Aat2 is polysome associated independent of the growth conditions.

To investigate whether polysome-associated Aat2 was primarily ribosome- or mRNA bound or part of another high-molecular-weight complex, we separated ribosome-bound proteins from free proteins on sucrose cushion gradients and analysed the pellet fraction on new polysome gradients

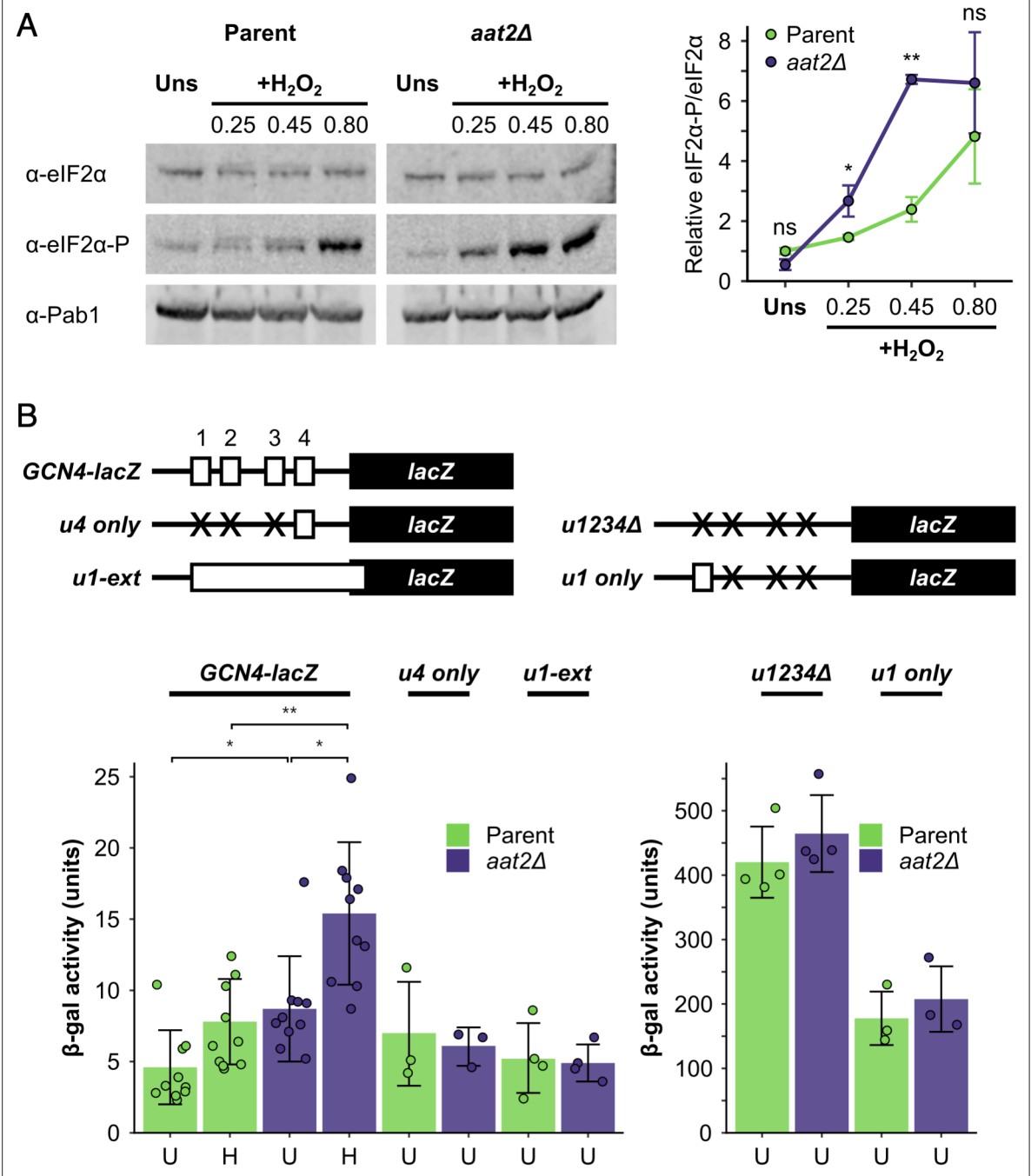

**Figure 5.** Deletion of *AAT2* enhances Gcn2 activity during oxidative stress. (**A**) Left: representative western blots showing eIF2α phosphorylation in unstressed and hydrogen peroxide ($H_2O_2$)-treated cultures. Right: bands were quantified using LI-COR Image Studio and the eIF2α-P/eIF2α ratio was calculated. Error bars show standard deviation (SD) ($n = 3$). (**B**) Top: *GCN4-lacZ* reporter constructs used to test the translational activation of *GCN4*. Solid boxes – *lacZ* ORF, open boxes – *GCN4* upstream ORFs, crosses – removed *GCN4* uORFs. Bottom: β-galactosidase activity in strains transformed with *GCN4-lacZ* reporter plasmids ($n = 3$–10). Error bars show SD. U – unstressed, H – +0.45 mM $H_2O_2$. The *t*-test was used to compare the strains: ns – not significant ($p > 0.05$), *$p < 0.05$, **$p < 0.01$.

with or without treatment with RNase I. In the absence of RNase I, Aat2 and control mRNA-binding proteins Pab1 and Scp160 were all polysome associated (*Figure 6B*). Following RNase I treatment, the mRNA-binding factors migrated at the top of the gradient, while Aat2 was predominantly 80S associated, indicating it is principally a ribosome-associated factor rather than an mRNA-binding protein (*Figure 6B*, lanes 5 and 6). This also ruled out the possibility that Aat2 was present in polysome

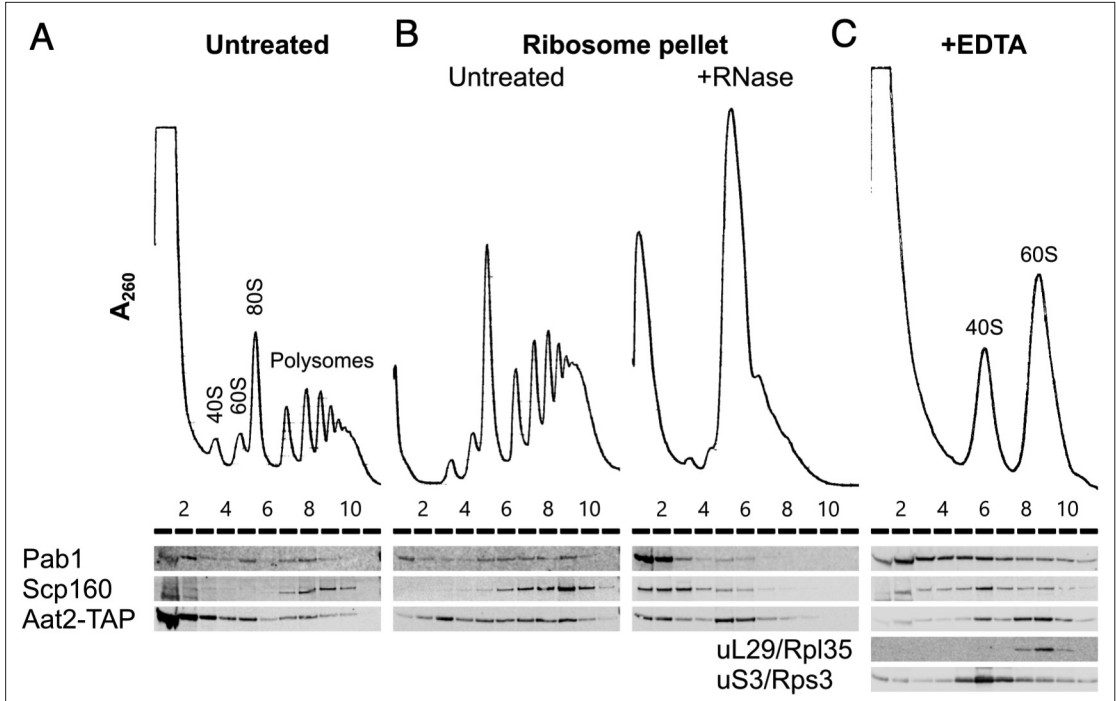

**Figure 6.** Aat2 binds to 60S ribosomes. (**A**) Representative polysome profile and western blot of an unstressed cell extract from the Aat2-TAP strain run on a 15–50% sucrose gradient. (**B**) Ribosome pellets from unstressed extracts were isolated using sucrose cushions, then either left untreated or treated with RNase I prior to polysome profiling. (**C**) Representative polysome profile and western blot from an unstressed cell extract treated with 50 mM ethylenediaminetetraacetic acid (EDTA) and run on a 10–25% sucrose gradient to separate the ribosomal subunits.

fractions through an association with another large cytoplasmic complex. EDTA treatment to separate the ribosomal subunits further indicated that Aat2 predominantly binds to 60S (large) subunits (*Figure 6C*, lanes 8–10).

## Aat2 ribosome engagement and signalling to Gcn2 is independent of aspartate aminotransferase function

Aat2 is a cytosolic aspartate aminotransferase enzyme that catalyses the reversible transamination reaction: 2-oxoglutarate + L-aspartate ↔ L-glutamate + oxaloacetate. By interconverting L-aspartate and oxaloacetate, the enzyme links amino acid and carbohydrate metabolism (*Figure 7—figure supplement 1A*). Aat2 has a paralog, Aat1, which performs the same reactions within mitochondria, although the cytosolic form is the major contributor of total aspartate aminotransferase activity (*Blank et al., 2005*). When grown on minimal medium, the *aat2Δ* mutant is an aspartic acid auxotroph, but when amino acids are supplied, as here in our experiments, cells grow well as Aat2 enzymatic function is not required (*Figure 7—figure supplement 2A*). To determine if the polysome-associated role of Aat2 in the stress response can be separated from its role in metabolism, we used a crystal structure of the Aat2 homodimer (PDB 1YAA, *Jeffery et al., 1998*) and its homology to other aspartate aminotransferases (*Winefield et al., 1995*) to identify active site residues K255 and R387, which engage the cofactor pyridoxal-5'-phosphate and the active site competitive inhibitor maleate, respectively (*Figure 7—figure supplement 1B*). We used a CRISPR/Cas9-mediated approach to introduce glutamic acid charge reversal mutations individually at these two positions in *AAT2-TAP* cells, as well as creating a control strain containing four silent mutations (SM) in the guide RNA-binding sequence. Importantly, both K255E and R387E mutations conferred aspartic acid auxotrophy (*Figure 7—figure supplement 2A*), despite being expressed well (*Figure 7—figure supplement 2B*). Both mutants retain ribosome-binding activity (*Figure 7—figure supplement 2C*). The mutations therefore compromise aspartate aminotransferase enzyme function, but not expression or ribosome binding.

All three versions of Aat2-TAP maintained the same polysome profile responses to oxidative stress (*Figure 7A*), unlike *aat2Δ* cells which show enhanced $H_2O_2$ sensitivity (*Figure 4C*), and the Aat2 mutant

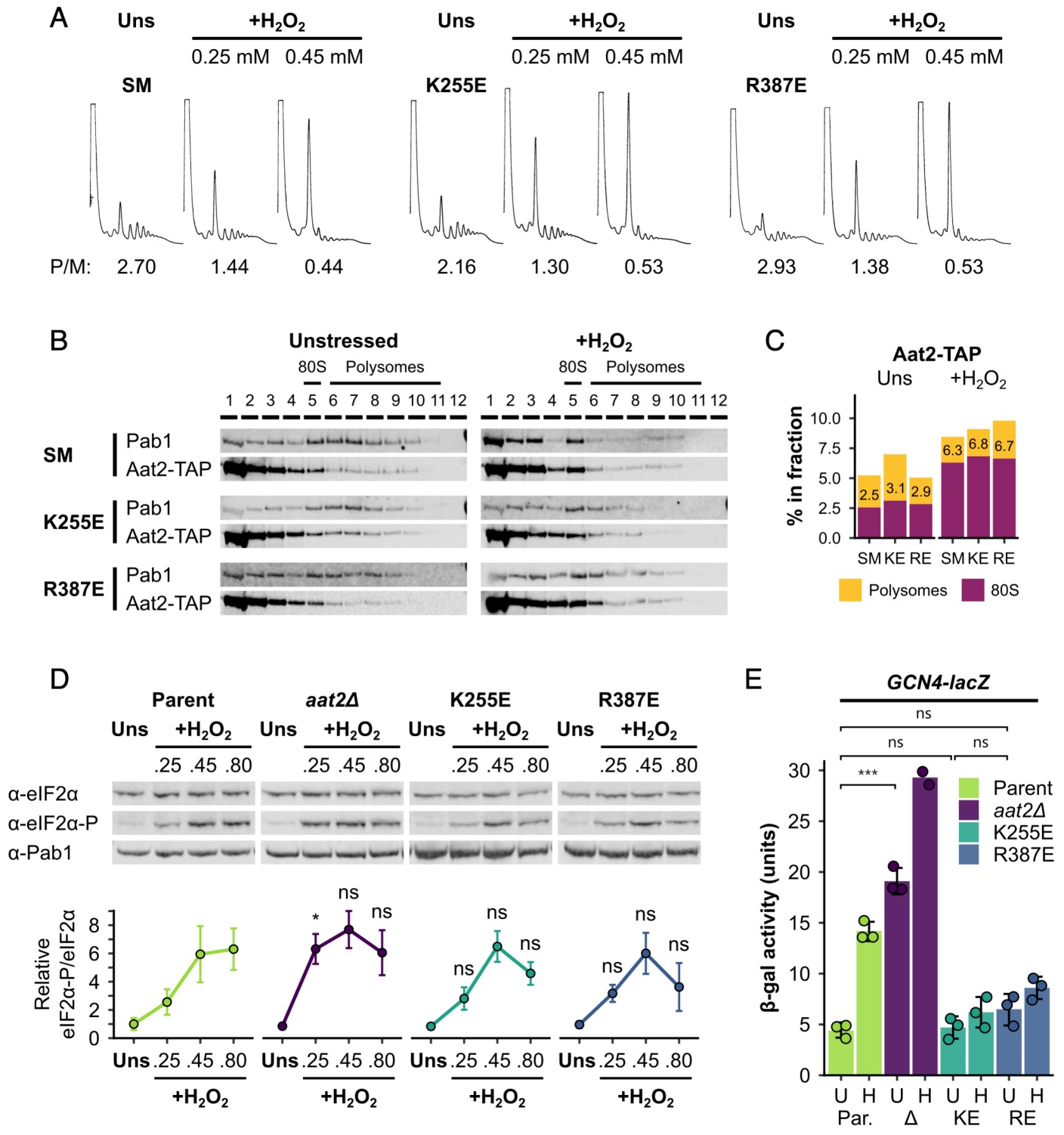

**Figure 7.** Non-catalytic mutants of Aat2 remain polysome associated and do not show heightened stress sensitivity. (**A**) Representative polysome profiles from unstressed and hydrogen peroxide ($H_2O_2$)-treated cultures. P/M: mean polysome-to-monosome ratio ($n$ = 2–3). (**B**) Representative western blots of sucrose gradient fractions from mutated Aat2-TAP strains. The positions of the 80S/monosome and polysome fractions are indicated. (**C**) Quantification of the 80S/monosomal and polysomal proportions of Aat2-TAP for each strain in (**B**). The monosomal percentage is indicated. (**D**) Top: representative western blots showing eIF2α phosphorylation in unstressed and $H_2O_2$-treated cultures. Bottom: bands were quantified using LI-COR Image Studio and the eIF2α-P/eIF2α ratio was calculated. Error bars show standard deviation (SD; $n$ = 3–4). (**E**) β-Galactosidase activity in strains transformed with *GCN4-lacZ* reporter plasmids ($n$ = 3). Error bars show SD. U – unstressed, H – +0.45 mM $H_2O_2$. The $t$-test was used for comparisons in

*Figure 7 continued on next page*

*Figure 7 continued*

(**D**) and (**E**): ns – not significant (p > 0.05), *p < 0.05, ***p < 0.001. In (**D**), each *t*-test result refers to the comparison with the equivalent condition in the parent strain.

The online version of this article includes the following figure supplement(s) for figure 7:

**Figure supplement 1.** Non-catalytic *AAT2* mutants are auxotrophic for aspartate and well expressed.

**Figure supplement 2.** Non-catalytic AAT2 mutants are auxotrophic for aspartate, well expressed and co-migrate with ribosomes.

**Figure supplement 3.** Silent mutations do not affect oxidative stress sensitivity or Aat2 polysome association.

proteins all migrated into polysomes in both unstressed and stressed cells (*Figure 7B, C*). The stress-dependent induction of eIF2α phosphorylation in the K255E and R387E mutants was normal, unlike the heightened response to low-level stress that was observed in *aat2Δ* cells (*Figure 7D*). These data suggest that binding of catalytically inactive Aat2 to ribosomes is sufficient to restore normal activation of Gcn2 and the global translational repression response to acute stress, unlike the hypersensitive response of *aat2Δ*. Despite this, *GCN4* induction in the mutants differed from both parent and *aat2Δ* cells. Both mutants showed normal *GCN4-lacZ* activity under unstressed conditions, but failed to induce *GCN4-lacZ* expression during oxidative stress (*Figure 7E*). This last observation points to a potential further role of Aat2 downstream of eIF2 phosphorylation. Taken together, these observations indicate that the ribosome/polysome-binding and aspartate aminotransferase enzyme functions of Aat2 are separable activities and that Aat2 can modulate the yeast ISR.

## Discussion
### Polysome proteomics

We used polysome profiling in combination with MS to survey proteins associated with the translational machinery. Our polysomal proteomics approach (*Figure 1*) captures many aspects of translation and its regulation during stress, including the behaviour of RPs, initiation factors, elongation factors, and RBPs (*Figures 2 and 3*). This is a powerful approach and is unbiased in the identification of proteins in sucrose gradient fractions, although we cannot definitively rule out the co-sedimentation of other large cytoplasmic assemblies with polysomes, for example P bodies or stress granules. Some of the proteins detected in ribosomal fractions therefore may not be truly RNA- or ribosome associated. However, all the proteins shown in *Figures 2 and 3C* were detected in ribosome fractions under unstressed conditions, when P bodies and stress granules are absent in yeast. Furthermore, the large enrichment of RNA-binding and translation-related proteins in our data gives confidence in polysomal proteomics to assess changes in proteins associated with the translational machinery during stress.

Our unbiased clustering reveals that proteins displaying common changes in polysome association profiles across unstressed and stressed conditions share common functions in translational control. These shared regulatory responses likely result from the reductions in growth rates and global translation initiation and elongation activity that occur (*Figure 1*). As expected, when global translation initiation was reduced, initiation factors reduced in relative ribosome association during stress. This applied to almost all, except for the 60S joining factor eIF5B and the RNA helicases eIF4A (along with its associated eIF4B) and Ded1 (*Figure 2*). The RNA helicases associated with polysomes equally in unstressed and stressed cells. This is perhaps surprising, because eIF4A, eIF4B, and Ded1 were found to be rapidly lost from polysomes following both glucose starvation and heat shock (*Castelli et al., 2011*; *Bresson et al., 2020*). Perhaps importantly, neither glucose withdrawal for 10–15 min nor the immediate response to heat shock requires eIF2α phosphorylation (*Ashe et al., 2000*; *Grousl et al., 2009*; *Castelli et al., 2011*). eIF4A has at least two separate roles in yeast translation: first, it contributes to the recruitment of the 43S pre-initiation complex (PIC) to the 5′ end of capped mRNAs, and second it helps to unwind mRNA secondary structures during scanning towards the AUG start codon (*Yourik et al., 2017*; *Merrick and Pavitt, 2018*). Here, the mRNA cap-binding factor eIF4E and its partner eIF4G were the most ribosome-depleted initiation factors during stress (*Figure 2*), while their global RNA binding was found to be unchanged following glucose starvation or heat stress (*Bresson et al., 2020*). It was also noted previously that protein–protein interactions among these 5′ cap-associated factors were not altered by stress, but that interactions with specific mRNAs were changed when these were quantified using a RNA-immunoprecipitation and RNA-sequencing

approach (*Costello et al., 2017*). Together these data are consistent with models where $H_2O_2$ and amino acid starvation act at early time points via enhanced eIF2α phosphorylation to reduce 43S PIC–mRNA association, while heat shock and glucose starvation instead retain these 43S PIC–mRNA interactions but act to impede the subsequent scanning step via selective loss of RNA helicases.

In addition to impacting translation initiation, both stresses we investigated impair translation elongation (*Shenton et al., 2006*; *Harding et al., 2019*; *Wu et al., 2020*). Stalled or collided ribosomes have been proposed to facilitate activation of Gcn2 (*Pochopien et al., 2021*). In this model of Gcn2 activation, a blockage in elongation (e.g. caused by a lack of available aminoacylated tRNA) logically precedes the rapid attenuation of initiation. Slower migration of ribosomes increases ribosome density (*Yu et al., 2015*). We observed increased interactions of both eEF2 (*EFT2*) and eIF5A (*HYP2*) with ribosomes during stress (*Figure 2*). eEF2 (cluster 2) promotes ribosomal translocation and its PE was maintained under all conditions (*Figure 3*). Enhanced eEF2 phosphorylation during oxidative stress (*Wu et al., 2020*) may reduce its activity and lead to the higher ribosome interaction we observe. In contrast, eIF5A (cluster 11, *Figure 3*) binds to the ribosomal exit site to promote peptide bond formation on stalled or paused 80S ribosomes, and has recently been observed in Gcn1-bound disome structures (*Schuller et al., 2017*; *Pochopien et al., 2021*). Thus, the increased association of eIF5A with ribosomes makes mechanistic sense in the context of stress-induced ribosome pauses during elongation.

The RPs generally behaved coherently in their ribosome association in our data, becoming highly enriched in the monosomal fraction (F1) during both oxidative stress and histidine starvation, consistent with most ribosomes remaining intact (*Figure 3*). However, differences in the degree to which some RPs became monosome-enriched during stress were observed. Rpl8b/uL8 and Rps10/eS10 (cluster 13, *Figure 3*) were both more depleted from the heavy polysome fractions (F4–F5) during stress than the other RPs, suggesting that they might be selectively depleted from the mRNA-engaged pool of ribosomes under these conditions. Rpl8 has been implicated in contributing to 60S biogenesis: the absence of Rpl8 causes the depletion of several large subunit RPs and ribosome assembly factors from pre-ribosomes, including Rpl28/uL15 (cluster 12) (*Jakovljevic et al., 2012*). Rps10 is located within the mRNA entry channel and contacts mRNA during translation. It is commonly mutated in Diamond–Blackfan Anemia, an inherited bone marrow failure syndrome (*Doherty et al., 2010*). A recent study showed that the ribosome assembly factor Ltv1 promotes the incorporation of Rps10, Rps3, and Asc1 into the 40S head and that *ltv1Δ* cells have growth defects, including enhanced sensitivity to oxidative stress (*Collins et al., 2018*). The mammalian ortholog of Rps10 is a target for ZNF598 (the homologue of Hel2) ubiquitination (*Garzia et al., 2017*). Yeast Rps10 can also be ubiquitinated (*Swaney et al., 2013*), so partial ubiquitination of Rps10 might explain its reduced detection by MS in heavy polysomes. The four other members of cluster 13 (Dbp2, Nmd3, Arb1, and Arx1) are involved in ribosome biogenesis, nonsense-mediated decay (NMD) or RQC (*Woolford and Baserga, 2013*), consistent with their strong depletion from heavy polysomes in both unstressed and stressed conditions.

Uniquely among RPs, Rpl7b/uL30 was placed in cluster 6 along with mRNA-binding factors such as Pab1, which remained polysome-enriched during both stresses. Isoform-specific roles have been investigated for this paralog pair, with potential differential impacts on Ty element transposition described (*Palumbo et al., 2017*). In our MS data, the signal for Rpl7a (thought to be the major isoform) differed from that for Rpl7b, peaking in F5 in unstressed cells and F2 (disome/trisomes) during both stresses (*Supplementary file 1* – sheet 2). Further dedicated studies will be needed to determine whether these RPs have specific roles in stress responses.

## Aat2 alters the sensitivity of the ISR to oxidative stress

Here, we identified Aat2 as a ribosome-binding factor modulating the oxidative stress sensitivity of yeast cells. We found that the protein was polysome associated in unstressed growth conditions and maintained its PE during both oxidative stress and amino acid starvation. Loss of Aat2 enhances both stress sensitivity and the activation of the eIF2 kinase Gcn2 (*Figures 4 and 5*). In yeast, Gcn2 is the sole eIF2 kinase activating the ISR (*Hinnebusch, 2005*; *Pavitt, 2018*). Previously, it was found that mutations deleting RPs or associated factors can modulate Gcn2 activation. For example, deletion of *RPS10A* or *RPS10B* (both encoding Rps10/eS10) limits the activation of Gcn2 kinase in replete conditions and following amino acid starvation (*Lee et al., 2015*), thus slowing activation of the ISR. Rps10 was found to contact Gcn1 in yeast two-hybrid experiments, suggesting that the RP can modify the

sensitivity of Gcn1, a factor which is necessary for Gcn2 activation (*Lee et al., 2015*). In contrast, loss of the RQC factor Hel2 promotes or enhances eIF2α phosphorylation both in unstressed cells and in response to the alkylating agent MMS (*Yan and Zaher, 2021*). Hel2 is an E3 ligase that ubiquitinates Rps20/uS10 and Rps3/uS3 (both cluster 12) in response to ribosome stalling to initiate RQC (*Matsuo et al., 2017*). Since RQC is inhibited in the absence of Hel2 and the Gcn2 activator Gcn1 binds to stalled/collided disomes, these findings suggested a model whereby Hel2 helps to resolve moderate stalls, and that Gcn2 activation ensues when Hel2 is unable to act or is overwhelmed during stress (*Yan and Zaher, 2021*). We observed heightened eIF2α phosphorylation at low $H_2O_2$ levels after only 15 min of treatment in *aat2Δ* cells (*Figure 4*), which resembles these recent *hel2Δ* observations under similar stress conditions. Aat2 (*Figure 6*) and Gcn2 (*Ramirez et al., 1991*; *Inglis et al., 2019*) both bind to 60S ribosomal subunits, suggesting a potential model where Aat2 binding to polysomal 60S subunits in optimal growth conditions can antagonise Gcn2 activation. At present we cannot say how direct this role of Aat2 is, only that it appears to be acting upstream of Gcn2. Aat2 functions in amino acid metabolism as one of two enzymes that interconvert aspartate and glutamate, so we performed our studies under conditions where amino acid supply is in excess and its aspartate aminotransferase function is not essential. Two mutants that target key residues for catalytic function remain able to bind translating ribosomes and do not show the aberrant eIF2-phosphorylation and polysome profile patterns in response to stress that are characteristic of *aat2Δ* cells (*Figure 7*). These results imply that the global ISR-repressing role of Aat2 in unstressed cells is distinct from its aspartate aminotransferase function. However, while the global stress-response signalling and activation of Gcn2 appears normal in the *aat2* catalytic mutants, *GCN4-lacZ* reporter expression was repressed. This suggests that there may a further role of Aat2 downstream of Gcn2 activation and eIF2 phosphorylation that is impacted in these strains. It was recently shown that Gcn2 can phosphorylate additional substrates relavent to the ISR including Gcn20 and the eIF2 beta subunit (*Dokládal et al., 2021*). Investigating if any of these or other downstream events necessary for *GCN4* translational activation are impacted by Aat2 will require additional tools beyond the scope of this study.

Enzymes with secondary roles have been termed 'moonlighting enzymes' (*Castello et al., 2015*) and those that bind RNA have been proposed to act as a post-transcriptional network linking metabolism to gene regulation (*Hentze and Preiss, 2010*). The best-studied example of an RNA-binding metabolic enzyme is IRP1. In the presence of iron, IRP1 functions independently of RNA as the enzyme aconitase, while in the absence of iron it instead binds to specific target mRNAs via stem loop structures known as iron response elements, thus regulating their stability or translation. Other examples from glycolysis and other metabolic pathways have been reviewed elsewhere (*Hentze and Preiss, 2010*). However, as far as we are aware, Aat2 is the first example of a metabolic enzyme that binds to ribosomes and modulates the ISR.

# Materials and methods

**Key resources table**

| Reagent type (species) or resource | Designation | Source or reference | Identifiers | Additional information |
|---|---|---|---|---|
| Antibody | α-Rps3 (rabbit polyclonal) | PMID:11278502 | | Diluted in 5% milk-TBST (1:5000) |
| Antibody | α-Rpl35 (rabbit polyclonal) | PMID:11278502 | | Diluted in 5% milk-TBST (1:5000) |
| Antibody | α-Pab1 (mouse monoclonal) | EnCor Biotechnology | RRID:AB_2572370 | Diluted in 5% milk-TBST (1:4000) |
| Antibody | α-Scp160 (rabbit polyclonal) | PMID:11278502 | | Diluted in 5% milk-TBST (1:10,000) |
| Antibody | α-protein A (rabbit polyclonal) | Sigma-Aldrich P1291 | RRID:AB_1079562 | Diluted in 5% milk-TBST (1:1000) |
| Antibody | α-Sui2 (chicken polyclonal) | PMID:31086188 | | Diluted in 5% milk-TBST (1:1000) |
| Antibody | α-phospho-Sui2 (rabbit polyclonal) | Cell Signalling Technologies | RRID:AB_330951 | Diluted in 5% milk-TBST (1:1000) |
| Antibody | IRDye 800CW Goat anti-Rabbit IgG, H + L (goat polyclonal) | LI-COR Biosciences | Cat. no.: 926-32211 | Diluted in 5% milk-TBST (1:10,000) |
| Antibody | IRDye 680RD Goat anti-Mouse IgG, H + L (goat polyclonal) | LI-COR Biosciences | Cat. no.: 926-68070 | Diluted in 5% milk-TBST (1:10,000) |

*Continued on next page*

*Continued*

| Reagent type (species) or resource | Designation | Source or reference | Identifiers | Additional information |
|---|---|---|---|---|
| Antibody | IRDye 680RD Donkey anti-Chicken Ig, H + L (donkey polyclonal) | LI-COR Biosciences | Cat. no.: 926-68075 | Diluted in 5% milk-TBST (1:10,000) |
| Chemical compound, drug | Hydrogen peroxide solution | Sigma | Cat. no.: 95,321 | Made 1/100 dilution in water before adding to cultures/media |
| Chemical compound, drug | 3-Amino-1,2,4-triazole | Sigma | Cat. no.: A8056 | |
| Peptide, recombinant protein | Ambion RNase I | Thermo Fisher | Cat. no.: AM2294 | |
| Peptide, recombinant protein | Sequencing Grade Modified Trypsin | Promega | Cat. no.: V5111 | |
| Commercial assay or kit | NuPAGE 4% to 12%, Bis-Tris, 1.0 mm, Mini Protein Gel, 12-well | Life Technologies | Cat. no.: NP0322 | |
| Strain, strain background (*E. coli*) | XL10-Gold Ultracompetent Cells | Agilent | Cat. no.: 200,314 | |
| Strain, strain background (*S. cerevisiae*) | S288c | Detailed in **Supplementary file 1**, Sheet 6 | | |
| Software, algorithm | LI-COR Image Studio and LI-COR Image Studio Lite | LI-COR Biosciences | https://www.licor.com/bio/software | Version 5.2. LI-COR Image Studio Lite has since been discontinued. |
| Software, algorithm | R | **R Core Team, 2019** | https://www.r-project.org/ | Version 3.6.2 |
| Software, algorithm | MaxQuant | **Tyanova et al., 2016** | https://www.maxquant.org/ | Version 1.5.7.4 |
| Software, algorithm | GNU Image Manipulation Program | GIMP | https://www.gimp.org/ | Version 2.8.10 |

## Yeast strains

Yeast strains used in this study are described in **Supplementary file 1** – sheet 6. Plasmids are described in **Supplementary file 1** – sheet 7. All strains used were from the BY4741 background of S288c. Strains made in this study were constructed using standard yeast transformation and site-directed mutagenesis using the CRISPR/Cas9 system (**Anand et al., 2017**). Oligonucleotides used to make mutations or confirm gene deletions or mutations are listed in **Supplementary file 1** – sheet 8.

## Cell growth conditions

*Saccharomyces cerevisiae* colonies were inoculated into 5 ml synthetic complete medium containing dextrose (2%) lacking histidine (SC –His) and grown overnight at 30°C with rotation at 180 rpm. The next day, 200 ml cultures were started at an $OD_{600}$ of 0.1. For oxidative stress experiments, $H_2O_2$ was added during exponential growth ($OD_{600}$ 0.6–1.0) at a final concentration of 0.2–0.8 mM. For histidine starvation, 3-AT) was added during exponential phase at a final concentration of 10 mM.

## Site-directed mutagenesis

Two point mutations were separately introduced into AAT2-TAP using the CRISPR/Cas9 system. A guide RNA targeting the *AAT2* coding sequence was designed using the ATUM web tool (https://www.atum.bio/eCommerce/cas9/input) and cloned into plasmid pAV2676, containing the *Cas9* gene under the control of the *PGK1* promoter (**Supplementary file 1** – sheet 7). Three repair oligonucleotides were designed to target repair to the *AAT2* locus and generate point mutants: control, K255E and R387E (**Supplementary file 1** – sheet 8). Each contained four silent mutations within the guide RNA target sequence to ensure the locus was not re-cut following repair, without changing the amino acid sequence. The *Cas9*/guide RNA plasmid and repair construct oligonucleotide were simultaneously transformed into the *AAT2-TAP* strain (GP7542) and transformants selected by plating on SC –Leu. Transformants were screened by sequencing (Eurofins Genomics) to confirm the presence of the correct mutations and the *Cas9*/guide RNA plasmid was removed by growth on SC medium.

## Serial dilution growth spotting assays

*S. cerevisiae* cultures were grown to exponential phase then diluted to $OD_{600}$ 0.1 in sterile water in a 96-well plate. A dilution series was made for each strain in sterile water to give cultures at $OD_{600}$ 0.1, 0.01, 0.001, and 0.0001. Two microliters of each culture were plated on to SC and SC + $H_2O_2$ (final $H_2O_2$ concentrations 0.4–2.0 mM) agar plates. Plates were incubated at 30 or 37°C and imaged after 48 hrs.

## Cell extract preparation

Cultures were grown at 30°C to an $OD_{600}$ of 0.6–1.0. Formaldehyde crosslinking was used to fix translating ribosomes and their associated factors on mRNA. Cultures were transferred into pre-chilled Falcon tubes containing formaldehyde (at a final concentration of 0.8% [vol/vol]) and a quarter volume of frozen media pellets. Samples were incubated on ice for 1 hr then excess formaldehyde was quenched by the addition of 2 M glycine (final concentration 80 mM). Cells were harvested by centrifugation, washed and lysed with acid-washed glass beads in 200 µl polysome lysis buffer (20 mM N-2-hydroxyethylpiperazine-N'-2-ethanesulfonic acid (HEPES) pH 7.4, 2 mM magnesium acetate, 100 mM potassium acetate, 0.5 mM dithiothreitol (DTT), 0.1% diethyl pyrocarbonate (DEPC)) for 20 s, seven times.

## Polysome profiling

Two $A_{260}$ units of lysate were layered on to 15–50% sucrose gradients prepared in 12 ml thin-walled open polyallomer tubes (Seton Scientific) and separated by ultracentrifugation in an SW41 Ti rotor (2.5 hr at 278,000 × $g$, 4°C). Profiles were generated by continuous $A_{254}$ recording using a UA-6 UV/Vis detector and chart recorder (Teledyne ISCO). Profile images were analysed using GNU Image Manipulation Program (version 2.8.10). Fractions were collected manually: for western blotting, 11–14 fractions (each 0.8–1 ml) were collected, starting from the top of the gradient; for MS, five fractions (each 1 ml) were collected, beginning at the start of the monosomal peak. For limited RNase digestion, 1 U/µl RNase I (Ambion) was added to lysates prior to polysome profiling and incubated for 30 min at room temperature. For EDTA treatment, 50 mM EDTA was added to lysates and incubated for 30 min on ice.

## Sucrose cushions

Lysates were prepared as for polysome profiling. Sixteen $A_{260}$ units of lysate (normalised to 500 µl volume) were layered without mixing on top of 400 µl of 36% sucrose in polysome lysis buffer, in thick-walled open polycarbonate tubes (Beckman Coulter) and separated by ultracentrifugation in a TLA120.2 rotor (Beckman Coulter; 1.5 hr at 278,000 × $g$, 4°C).

For polysome profiling, supernatants were concentrated in Amicon Ultra 3 kDa MW cut-off centrifugal concentrators (Millipore) and ribosome pellets were resuspended in polysome lysis buffer. For western blotting, protein was extracted from supernatants and pellets were resuspended directly in protein loading buffer (2× NuPAGE LDS sample buffer [Invitrogen], 715 mM 2-mercaptoethanol). Samples were incubated for 5 min at 95°C.

## Protein extraction

Protein was extracted from sucrose gradient fractions for analysis by western blotting and MS. A half volume of ice-cold 40% trichloroacetic acid (TCA) was added to each fraction, mixed by inversion and incubated overnight at –20°C. Precipitated protein was pelleted by centrifugation (15 min at 20,000 × $g$, 4°C) and washed twice with ice-cold acetone. Pellets were dried for 20 min at room temperature then resuspended in 5 µl of 1 M Tris and 10 µl of protein loading buffer (2× NuPAGE LDS sample buffer [Invitrogen], 715 mM 2-mercaptoethanol). Samples were incubated for 5 min at 95°C.

## SDS–PAGE and western blotting

Protein samples were resolved on NuPAGE 4–12% Bis-Tris gels (Invitrogen) and transferred to nitrocellulose membranes. Membranes were probed with monoclonal and polyclonal antibodies listed in *Supplementary file 1* – sheet 9 and visualised using LI-COR fluorescent secondary antibodies. Bands were quantified using LI-COR Image Studio (version 5.2).

## Label-free MS

Cytoplasmic extracts (referred to as Totals/T) were prepared as in 'Cell extract preparation'. These were loaded on to sucrose gradients, fractions collected, and protein extracted as in 'Polysome profiling' and 'Protein extraction'. Totals were prepared for MS by adding equal volumes of sample and protein loading buffer (2× NuPAGE LDS sample buffer [Invitrogen], 715 mM 2-mercaptoethanol), and incubating for 5 min at 95°C. Both Totals and Fractions were briefly run on NuPAGE 4–12% Bis-Tris gels (Invitrogen) and protein samples were excised from the gels. Samples were dehydrated using acetonitrile and centrifuged under vacuum. Dried gel pieces were reduced with 10 mM DTT and alkylated with 55 mM iodoacetamide, then twice washed alternately with 25 mM ammonium bicarbonate and acetonitrile. Gel pieces were dried by vacuum centrifugation and samples digested using trypsin overnight at 37°C.

Liquid chromatography was carried out using an UltiMate 3000 Rapid Separation Binary System (Thermo Fisher Scientific). Peptides were concentrated using an ACQUITY UPLC M-Class Symmetry C18 Trap Column (180 μm inner diameter, 20 mm length [Waters]) and then separated using an ACQUITY UPLC M-Class Peptide BEH C18 Column (75 μm inner diameter, 250 mm length, 1.7 μm particle size [Waters]). A gradient starting with 99% Buffer A (0.1% formic acid in water) and 1% Buffer B (0.1% formic acid in acetonitrile) and increasing to 75% Buffer A and 25% Buffer B was used to separate the peptides over 45 min at a flow rate of 200 nl/min. Label-free tandem MS was performed using an Orbitrap Elite Hybrid Ion Trap-Orbitrap Mass Spectrometer (Thermo Fisher Scientific). Peptides were selected for fragmentation and MS2 analysis automatically by data-dependent analysis.

## MS data processing

### Database searches

Raw MS data were processed using MaxQuant version 1.5.7.4 (*Tyanova et al., 2016*). A peptide mass tolerance of 20 ppm was used for the first search, 4.5 ppm for the main search, and 0.5 Da for the MS/MS fragment ions. The peak list was searched against the Uniprot *Saccharomyces cerevisiae* database (accessed 10 February 2017) using the built-in Andromeda search engine (*Cox et al., 2011*). Peptide-spectrum matches and protein groups were each filtered at a false discovery rate of 1%. 901 protein groups were identified in two or more replicates for at least one sample (F1–F5 or T, *Supplementary file 1* – sheet 1). Of these, 840 were uniquely identified proteins encoded by single genes, while 61 are 'grouped' proteins that are not distinguishable by MS (most of which are RP paralog pairs). Throughout the text 'protein' is used to refer to both grouped and uniquely identified proteins.

### Polysome enrichment profiles

Processed label-free data were analysed and presented using R (version 3.6.2) (*R Core Team, 2019*) and additional packages therein. LFQ intensity values from MaxQuant were used as the primary quantitative signal. To generate 'polysome enrichment profiles', LFQ intensities were $\log_2$ transformed and normalised for each protein to its mean across the unstressed ribosome fractions (F1–F5).

### Ribosome association

The overall ribosomal engagement of each protein was estimated by first summing the raw LFQ intensities in the ribosomal fractions (F1–F5) to calculate the fraction sum (FS) for each condition, then calculating the ratio between this and the respective Total (FS/T, $\log_2$ transformed). The proportion of each protein that is ribosome associated was estimated by inverse $\log_2$ transforming the FS/T ratio and dividing by 30 (to account for the difference in the relative amounts of the Fractions and Totals analysed by MS). Changes in ribosome association during stress were assessed by calculating the difference in FS/T from the condition of interest to unstressed (ΔFS/T). The RPs in general showed no change in overall ribosome association so were used to model a 'null distribution' of non-changing proteins (a normal distribution with the same ΔFS/T mean and standard deviation as the RPs), which was used to generate p values for each protein's ΔFS/T value.

### Hierarchical clustering

Hierarchical clustering was used to identify sets of proteins with similar 'polysome association profiles' using the R functions *dist* (method: 'euclidean') and *hclust* (method: 'complete').

## Peptide intensity distributions

To assess the contribution of nascent peptides to MS signal, cumulative peptide intensity from N- to C-terminus was calculated for proteins with a greater than median number of peptides detected (median = 7 peptides) and sequence coverage (proportion of residues that were detected in at least one peptide, median = 21.4%). Cumulative peptide intensity distributions from sucrose gradient fractions were compared with those from totals using the Kolmogorov–Smirnov test (with Bonferroni correction).

## β-Galactosidase assays

β-Galactosidase activity was measured in strains transformed with *GCN4-lacZ* plasmids to assay the translational induction of *GCN4* during oxidative stress as described (*Dever, 1997*). Strains were grown to exponential phase in 50 ml SC –uracil. Cells were harvested by centrifugation, washed and lysed with acid-washed glass beads in 200 µl of breaking buffer (0.1 M Tris pH 8, 20% glycerol, 1 mM 2-mercaptoethanol) for 20 s, five times. Another 200 µl of breaking buffer was added and mixed by vortexing. Samples were cleared by centrifugation to remove cell debris (10 min at 7000 × *g*, 4°C). β-Galactosidase activity was measured for 100 µl lysate mixed with 900 µl of Z-buffer (60 mM sodium phosphate dibasic heptahydrate, 45 mM sodium phosphate monobasic, 10 mM potassium chloride, 2 mM magnesium sulphate, and 40 mM 2-mercaptoethanol). Samples were incubated for 10 min at 28°C and reactions started by adding 200 µl ONPG (4 mg/ml in Z-buffer). Reactions were stopped by the addition of 500 µl 1 M sodium carbonate and $A_{405}$ was measured.

## Acknowledgements

We thank David Knight and colleagues at the University of Manchester BioMS Core Facility (RRID:SCR_020987) for processing mass spectrometry samples and their helpful advice, and Martin Jennings and Chris Kershaw for helpful advice and feedback throughout the study.

## Additional information

### Funding

| Funder | Grant reference number | Author |
| --- | --- | --- |
| Biotechnology and Biological Sciences Research Council | BB/M011208/1 | Robert A Crawford |
| Weizmann UK | 2020/129488 | Graham D Pavitt |

The funders had no role in study design, data collection, and interpretation, or the decision to submit the work for publication.

### Author contributions

Robert A Crawford, Investigation, Methodology, Project administration, Writing - original draft, Writing – review and editing; Mark P Ashe, Simon J Hubbard, Conceptualization, Supervision, Writing – review and editing; Graham D Pavitt, Conceptualization, Data curation, Formal analysis, Funding acquisition, Project administration, Resources, Supervision, Writing - original draft, Writing – review and editing

### Author ORCIDs

Robert A Crawford http://orcid.org/0000-0002-9788-5137
Mark P Ashe http://orcid.org/0000-0002-4457-7851
Simon J Hubbard http://orcid.org/0000-0002-8601-9524
Graham D Pavitt http://orcid.org/0000-0002-8593-2418

### Decision letter and Author response

Decision letter https://doi.org/10.7554/eLife.73466.sa1
Author response https://doi.org/10.7554/eLife.73466.sa2

## Additional files

### Supplementary files
• Transparent reporting form
• Source data 1. Uncropped Western blot images.

• Supplementary file 1. MS data analyses.

### Data availability
The mass spectrometry proteomics data have been deposited to the ProteomeXchange Consortium via the PRIDE (Perez-Riverol et al. 2019) partner repository with the dataset identifier PXD027903.

The following dataset was generated:

| Author(s) | Year | Dataset title | Dataset URL | Database and Identifier |
|---|---|---|---|---|
| Crawford RA, Ashe MP, Hubbard SJ, Pavitt GD | 2022 | Identification of polysome-associated translational regulators during oxidative stress and amino acid starvation | https://www.ebi.ac.uk/pride/archive/projects/PXD027903 | PRIDE, PXD027903 |

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
