## [Editor Report]

In this article, Crawford et al. monitored stress-induced alteration in proteins associated with translating ribosomes in yeast using mass spectrometry-based approaches. This revealed that cytosolic aspartate aminotransferase 2 (Aat2) is associated with polysomes. Aat2 deletion sensitizes yeast to oxidative stress which is paralleled by aberrantly elevated integrated stress response. The authors also show that polysome-association of Aat2 and its role in oxidative stress response are independent of its aminotransferase activity. Altogether, it was found that this study is of broad interest since it provides further evidence that metabolic enzymes may "moonlight" as post-transcriptional regulators while highlighting previously unappreciated aspects of adaptation to stress.

---

## [Decision Letter]

**Decision letter after peer review:**

Thank you for submitting your article "Cytosolic aspartate aminotransferase moonlights as a ribosome binding modulator of Gcn2 activity during oxidative stress" for consideration by *eLife*. Your article has been reviewed by 3 peer reviewers, including Ivan Topisirovic as the Reviewing Editor and Reviewer #1, and the evaluation has been overseen by a Reviewing Editor and David Ron as the Senior Editor. The following individual involved in review of your submission has agreed to reveal their identity: Leos Shivaya Valasek (Reviewer #3).

Essential revisions:

1) Some potential technical issues were observed pertinent to the mass spectrometry data including those related to diffusion and/or contamination with stress-induced cytoplasmic bodies that may co-sediment with polysomes. It was thus thought that the further evidence is warranted to confirm that Aat2 associates and not just simply co-sediments with ribosomes. This could be achieved e.g., by crosslinking-immunoprecipitation approaches or by performing in vitro ribosome binding assay (e.g., PMID: 18557705).

2) Although it was appreciated that elucidating precise mechanism underpinning stress-induced association of Aat2 with the polysomes and/or the effects of Aat2 on integrated stress response is outside of the scope of the present article, it was thought that further characterization of non-catalytic Aat2 mutants is warranted. To this end, the effects of non-catalytic Aat2 mutants on stress response should be further characterized by monitoring phosphorylation of eIF2alpha and the GCN4-lacZ activity.

3) Description of statistical methods used to analyze mass spectrometry data was found to be incomplete. Potential limitations of the polysome profiling/mass spectrometry approach should also be discussed. Finally, it was thought that a list of top hits should be provided to the readers.

*Reviewer #1 (Recommendations for the authors):*

– In general, it was found that there is a lack of mechanistic detail pertinent to Aat2 association with ribosomes. To this end, the conclusion that Aat2 associates with 60S ribosome is largely derived from their co-sedimentation presented in figure 6. However, at least as far as I could tell, association between Aat2 and 60S ribosomal subunit has not be demonstrated. It also remains unclear whether this is direct binding or indirect association. Although it is appreciated that addressing precise mechanisms of Aat2:ribosome association is likely to be out of the scope of the present study, at least some mechanistic insights appear to be warranted.

– It appears that the factors that mediate the signals from stressors to stimulate Aat2:ribosome association remain elusive. Does association of Aat2 with the ribosome require ongoing translation or not? Dynamics of Aat2:ribosome association are also not clear. Is Aat2 retained on the ribosomes throughout the adaptive phase? Does it dissociate during recovery from stress? Addressing some of these points was thought to be required to strengthen the proposed model.

– Notwithstanding proposed role of Gcn2, some additional mechanistic insights in how Aat2 moderated ISR would be appreciated.

– In figure 2B, eIF4E appears to be highly associated with polysomes. Is this a consequence of using FA over CHX? In other words, are the FA crosslinked 48S PICs present in polysomal fractions? Is there diffusion of PICs to disome fraction? Considering PMID: 31534220, it is also somewhat surprising that ~50% of eIF5B appears to be polysome associated. Finally, it appears that the authors were not able to validate 3 out of 4 hits from mass spec. Based on this, it may be advisable that the authors point out the shortcomings of their and similar approaches. Of note, in comparison to some similar studies, it was thought that his should not be a major reason of concern, considering that the authors validated Aat2 using elegant genetic and biochemical studies.

*Reviewer #2 (Recommendations for the authors):*

1. Define more fully in the legend the y-axis of graph (right side) of supplemental figure 1A – % in fraction.

2. page 7- Line 191: 465 proteins were detected in all of F1-F5, and 516 in all of F1-F4. Does this mean this number of proteins were identified to be in common in each of fractions 1 -5 or between 1-4? (ie this is not an aggregate number for this combination of fractions?). Line 209 polysome fractions appear to be referring to F1-5, which includes monosomes, di, tri-polysomes up to heavy polysomes (F5). Totals are presumed to be materials loaded onto the gradient, but this is not clear and not detailed in the methods.

3. Figure 2: The figure 1A and B y-axes emphasizes % in F1-F5, while the legend indicates in F1-F4. Is F1 a polysomal fraction (monosome in Figure 1)? Operational definitions for M and P need to be clear here and in later figures.

4. In the MS processing methods, it should be clear how statistical significance for changes in protein fraction distributions were determined.

*Reviewer #3 (Recommendations for the authors):*

It is a very interesting, topical, and well-executed work that reads very well and, in my opinion, meets the publishing standards of the *eLife* journal. It brings yet another example of until recently totally unimaginable players in translational control; i.e. metabolic enzymes. The manuscript will be of interest for all enthusiasts in the regulation of gene expression field. Below I list a few issues that might deserve the author's attention.

1) Can the authors fully rule out the cross-contamination of their polysomal fractions with various stress-induced large cytoplasmic assemblies / bodies? A definitive proof of principle experiment could be to tag one of the ribosomal proteins, IP the HCHO x-linked complexes containing it, reanalyze the purified samples with MS and compare the output with no-IP data. Performing this kind of an experiment, which is easy to do in yeast, perhaps only with the most critical part of the presented data, might dismiss all doubts and, perhaps, help to explain some inconsistencies in their PE proteomics data.

2) Figure 7. To further strengthen the argument that non-catalytic AAT2 mutants did not show heightened stress sensitivity, the authors should demonstrate that eIF2-α is not phosphorylated in these mutants (as in Figure 5A for the aat2delta strain) and measure the GCN4-lacZ (p180) activity too (as in Figure 5B).

---

## [Author Response]

Essential revisions:1) Some potential technical issues were observed pertinent to the mass spectrometry data including those related to diffusion and/or contamination with stress-induced cytoplasmic bodies that may co-sediment with polysomes. It was thus thought that the further evidence is warranted to confirm that Aat2 associates and not just simply co-sediments with ribosomes. This could be achieved e.g., by crosslinking-immunoprecipitation approaches or by performing in vitro ribosome binding assay (e.g., PMID: 18557705).

We appreciate the reviewers’ suggestions here and have attempted to address this issue in several ways.

Firstly, our data shows that a consistent fraction of Aat2 associates with polysomes both in the absence of, and the presence of, stress (Figure 3C). Individual comments by reviewers suggest that this point was not made clearly enough in our original manuscript as reviewers apparently thought Aat2 association with polysomes was exclusively promoted by stress. Hence, the observed constitutive association with polysomes would appear to rule out Aat2 association with ‘stress-associated cytoplasmic bodies’ which are not found in unstressed cells.

We have modified the text on page 10 to clarify:

'Aat2 was placed in cluster 3 in our polysomal proteomics, indicating that it is associated evenly throughout the polysome fractions in both unstressed and stressed cells, similar to the profiles for eIF4A and translation elongation factors (Figure 3C).'

Secondly, our experimental time window for examining the effect of stress on polysome-associated proteins is very short (15 mins). This is earlier than when P bodies and/or Stress granules typically appear during oxidative stress. Early in the study we undertook a microscopic analysis of P body and Stress granule formation using a strain with a marker for each (Dcp2-CFP for P bodies and Cdc33-RFP for stress granules). With this strain glucose starvation (which targets eIF4A) rapidly induced both types of granules in a high percentage of cells. However, the stresses we used in our study did not induce stress granules and especially oxidative stress was very slow at forming P-bodies such that at the earliest time possible for microscopy (25 mins) almost no P-bodies formed. Thus, we concluded that at our early time point during stress adaptation, protein synthesis initiation was impacted (polysome run-off) without causing appreciable formation of P-bodies and or stress granules which could interfere with our MS study.

Thirdly, studies aimed at determining P-body and stress granule composition have not identified Aat2 as part of these stress-induced cytoplasmic bodies (eg doi.org/10.1016/j.cell.2015.12.038 and DOI: 10.1080/15476286.2021.1976986). These bodies are also significantly larger than polysomes and are likely completely sedimented during centrifugation.

In summary it is highly unlikely that Aat2 co-sediments with stress induced P-bodies or stress granules. However, we acknowledge that there may be other high-molecular weight complexes that are not polysomes that Aat2 could potentially associate with, that these arguments cannot exclude, so we set out to address this experimentally.

We have attempted to address the point directly with an IP experiment. However, we were hampered by the lack of specific Aat2 antiserum, so are reliant on tagged Aat2 to detect it (except via MS). We used our TAP-tagged Aat2 strain to immunoprecipitate Aat2 from RNase-treated free and 80S/monosome fractions. Unfortunately, we encountered a common technical issue with IPs, which is that ribosomes often co-sediment non-specifically with the immunoprecipitating resin. To counter this, we applied a stringent wash with a high salt buffer and detergent which inevitably reduced the signals we were aiming to capture. We were able to see a modest enrichment of the ribosomal marker protein uS3/Rps3 in 3 of 4 replicates done (mean enrichment of Aat2-TAP strain over parent=1.630±0.575). Hence, although we feel that this experiment does confirm Aat2-ribosome interaction via an alternative technique, it does not enhance the overall message of our paper so have not included it in the main manuscript.

In summary, the technical complications described here prevented us from completing what we thought would be relatively straightforward experiments. Hence, we have modified the text at the start of the Discussion section to indicate it remains possible that some of the proteins identified may co-sediment rather than co-associate with polysomes.

2) Although it was appreciated that elucidating precise mechanism underpinning stress-induced association of Aat2 with the polysomes and/or the effects of Aat2 on integrated stress response is outside of the scope of the present article, it was thought that further characterization of non-catalytic Aat2 mutants is warranted. To this end, the effects of non-catalytic Aat2 mutants on stress response should be further characterized by monitoring phosphorylation of eIF2alpha and the GCN4-lacZ activity.

We have now completed these experiments and they are included in new panels in a revised Figure 7 – panels D and E. They show that the mutants (which are enzyme auxotrophs, but retain ribosome association) exhibit a normal eIF2 phosphorylation response unlike *aat2∆* cells (Figure 7D). They lower *GCN4* expression under optimal growth but do not induce Gcn4 expression upon the extended oxidative stress needed to induce *lacZ* expression in WT cells (Figure 7E). The mutants therefore behave distinctly to the *aat2∆* strain and support the idea that catalytic aminotransferase enzyme function and the protein’s role in the ISR are distinct.

3) Description of statistical methods used to analyze mass spectrometry data was found to be incomplete. Potential limitations of the polysome profiling/mass spectrometry approach should also be discussed. Finally, it was thought that a list of top hits should be provided to the readers.

We have amended the Methods to include more details on the processing of the MS data and have added an additional figure supplement to support Figure 2. A list of top hits has been included within Supplementary file 1. We have also discussed potential limitations of the experimental approach in the Discussion. Each point is discussed in further detail below.

Firstly, we have added in further details about the statistical processing within MaxQuant (Methods, page 21):

“The peak list was searched against the Uniprot *Saccharomyces cerevisiae* database (accessed 10^th^ February 2017) using the built-in Andromeda search engine (Cox et al. 2011). Peptide-spectrum matches (PSMs) and protein groups were each filtered at a false-discovery rate (FDR) of 1%…”

Second, we have added additional data processing for the changes in FS/T (% ribosome association) during stress, shown in Figure 2B and Figure 2—figure supplement 1. We assessed the statistical significance of changes in FS/T for proteins during each stress condition by modelling a population of proteins that does not change in FS/T compared with unstressed conditions. We assumed that the ribosomal proteins (RPs) did not generally change in their % ribosome association during stress, which was confirmed by both the SDS-PAGE and MS data, so we used this as a normally distributed sample sub-population against which to assess other proteins and estimate p-values (see dotted line indicating ‘null’ distribution in new Figure 2 —figure supplement 1B). The arrows on the right of Figure 2B have been updated to reflect this, now showing only statistically significant changes (p < 0.05) in FS/T. These data are now shown in Supplementary file 1–Sheet 3. The Methods have been updated to reflect the additional data processing.

Third, we have included a list of top hits in Supplementary file 1–Sheet 4. Proteins were defined as ‘top hits’ from the data presented in Supplementary file 1–Sheet 3 based on three criteria:

(1) Estimated percentage ribosome association in unstressed conditions. Only candidates with an estimated % ribosomal greater than 5% were considered.

(2) Change in ribosome association during H2O2 and/or 3-AT stress. Only candidates showing significant changes in ribosome association during one or both stresses were considered (i.e. ΔFS/T p-value < 0.05).

(3) Known function. RPs and TFs (translation factors) were excluded from the list of candidates, as the focus of the study was to identify proteins with previously under-appreciated roles in translational regulation.

The 74 candidate proteins defined by these criteria were then assigned a rank. Proteins showing significant changes during both stress conditions were ranked above those showing changes in only one, and proteins were then ranked based on their estimated % ribosome association (highest to lowest). Although not exhaustive, we believe that this list highlights many of the most interesting candidates for further investigation.

Fourth, we investigated methods to assess the statistical significance of the clustering shown in Figure 3C. We note that in the wider literature the statistical significance of clustering is not routinely tested, with unsupervised clustering methods instead being used to gain a broad overview of interesting patterns within the data. We did attempt to apply the methodology implemented in the R package *sigclust2* (Kimes *et al.,* 2017 PMID: 28099990) to analyse our clusters. *Sigclust2* tests whether the two groups (‘clusters’) separated at each node of the dendrogram are likely to belong to different normally distributed populations or are subsets of the same normally distributed population. It does this hierarchically, working down from the top of the clustering tree so that each node is only tested if its ‘parent’ showed a significant separation between the ‘daughter’ clusters.

We analysed our data (that used in Figure 3C) using *sigclust2*, however we encountered issues that prevented us from analysing the data fully in this way. Notably, *sigclust2* does not accept missing values, in contrast to the base R functions *dist* and *hclust*, which we used originally for our clustering. Only five proteins (of 353) contained one to three missing values (only in the Totals), while the rest contained none, but these still affect the clustering so *sigclust2* did not reproduce the exact same pattern as our original base R method – not a surprising result given the sensitivies of most clustering algorithms to minor variations in parameters or data. When comparing the two results however, the major divisions in the dendrogram remained and were significant (including the major division between clusters 1-3 and 4-14, and between clusters 6-11 and 12-14; p < 0.05). Given the strong similarity in the cluster sets derived in the original paper with the *sigclust2* set, with the latter offering statistical significance on the main separations, this persuaded us that the current clustering represents groups with distinctly different profiles and hence biologically significant patterns. We have therefore not replaced the clusters or figures in the current manuscript.

Reviewer #1 (Recommendations for the authors):– In general, it was found that there is a lack of mechanistic detail pertinent to Aat2 association with ribosomes. To this end, the conclusion that Aat2 associates with 60S ribosome is largely derived from their co-sedimentation presented in figure 6. However, at least as far as I could tell, association between Aat2 and 60S ribosomal subunit has not be demonstrated. It also remains unclear whether this is direct binding or indirect association. Although it is appreciated that addressing precise mechanisms of Aat2:ribosome association is likely to be out of the scope of the present study, at least some mechanistic insights appear to be warranted.

This point is addressed in the response to Essential Revisions point 1. We attempted to address this question by immunoprecipitating Aat2-TAP from both RNase-treated free and 80S/monosomal fractions and probing for ribosomal proteins in the resulting western blots. However, we found that ribosomal proteins non-specifically co-sedimented with the immunoprecipitating resin, which necessitated very stringent washing that reduced the signal of interest. We observed a modest enrichment of uS3/Rps3 in the IP fraction compared with the untagged control on average (mean=1.630 ±0.575, n=4), although this was not statistically significant (*p*=0.071, t-test, Reviewer Figure 2).

– It appears that the factors that mediate the signals from stressors to stimulate Aat2:ribosome association remain elusive. Does association of Aat2 with the ribosome require ongoing translation or not? Dynamics of Aat2:ribosome association are also not clear. Is Aat2 retained on the ribosomes throughout the adaptive phase? Does it dissociate during recovery from stress? Addressing some of these points was thought to be required to strengthen the proposed model.

Aat2 is present in monosomal and polysomal fractions in unstressed conditions as well as during stress, in both MS data (Figure 3C) and western blot data using Aat2-TAP (Figure 6A and Figure 7B).

There appears to have been some confusion with the interpretation of the polysome proteomics data so we have reworded some parts of the Results sections to make this point clearer. We have amended the description of the metabolic enzyme clusters (page 10) to reflect that these proteins are polysome-associated under unstressed conditions as well as during stress:

“Cytosolic aspartate aminotransferase moderates the ISR

Gene ontology analysis revealed that many of the most highly polysome-enriched proteins were metabolic enzymes, including glycolytic enzymes (e.g. Pgk1, Fba1, Eno1) and amino acid biosynthetic enzymes (e.g. Met6, His4, Trp5) (Figure 3C, Figure 3—figure supplement 1, Supplementary file 1–sheet 4). These proteins are polysome-associated under unstressed conditions and retain or increase both their ribosome association and PE during stress (clusters 1-6 and 8-10). While their association with mRNA is known (Supplementary file 1–sheet 3), their additional roles (if any) remain unclear …”

We have also stressed this point further with respect to Aat2 specifically (page 10/11), to clarify that it is present in F1-F5 in both unstressed and stressed cells (and therefore that its association is not dramatically altered by stress):

“Aat2 was present in cluster 3 in our polysomal proteomics, indicating that it is associated evenly throughout the polysome fractions in both unstressed and stressed cells, similar to eIF4A and elongation factors (Figure 3C).”

Finally, we stressed this point again in the Discussion:

“Aat2 alters the sensitivity of the ISR to oxidative stress

Here we identified Aat2 as a ribosome binding factor modulating the oxidative stress sensitivity of yeast cells. We found that the protein was polysome-associated under unstressed conditions and maintained its polysome enrichment during both oxidative stress and amino acid starvation …”

– Notwithstanding proposed role of Gcn2, some additional mechanistic insights in how Aat2 moderated ISR would be appreciated.

This point is addressed in the response to Essential Revisions point 2. We have included new experiments measuring eIF2α phosphorylation and *GCN4-lacZ* expression in both of our point mutant strains (revised Figure 7, panels D and E). Neither mutant strain has an altered pattern of eIF2α phosphorylation, unlike *aat2Δ* (Figure 7D). In addition, both mutants show normal *GCN4-lacZ* expression in unstressed cells, but do not activate *GCN4-lacZ* translation during oxidative stress (Figure 7E).

– In figure 2B, eIF4E appears to be highly associated with polysomes. Is this a consequence of using FA over CHX? In other words, are the FA crosslinked 48S PICs present in polysomal fractions? Is there diffusion of PICs to disome fraction? Considering PMID: 31534220, it is also somewhat surprising that ~50% of eIF5B appears to be polysome associated.

The current understanding of the mechanism of initiation is that it should occur on both free mRNAs (i.e. where the initiation event is the first event/pioneer round on that mRNA) and mRNAs that already contain elongating ribosomes, otherwise no polysomes should form. Therefore we would expect that initiation factors would associate with polysomal complexes (as well as monosomes). It is not thought that any initiation factors are only required for the first initiation event. Our view is that the associations of those factors shown in Figure 2B are entirely consistent with current ideas of how initiation operates.

FA vs CHX: We used FA in order to stabilise complexes to retain RPs, TFs, RBPs and other proteins during ultracentrifugation. CHX would only retain ribosomes on the mRNA, whereas other proteins would potentially be lost if they detach from polysomes during ultracentrifugation, causing them to migrate to lighter sucrose concentrations. FA is widely used to stabilize initiation complexes. Combined with the above logic, the data are consistent with FA-crosslinked 48S PICs being present in polysomal fractions.

Finally, it appears that the authors were not able to validate 3 out of 4 hits from mass spec. Based on this, it may be advisable that the authors point out the shortcomings of their and similar approaches. Of note, in comparison to some similar studies, it was thought that his should not be a major reason of concern, considering that the authors validated Aat2 using elegant genetic and biochemical studies.

We are unsure where the statement “the authors were not able to validate 3 out of 4 hits from mass spec” comes from. We used western blotting with both antibodies (to Rps3, Rpl35, eIF4G, eIF4A, eIF4E, Pab1 (shown in Figure 6, left panel) and Scp160 (shown in Figure 6, left panel)) and TAP-tagged candidate strains (Aat2 (shown in Figure 6, left panel), Clu1, Rbg1, Bmh1, Cdc19, Psa1, Met6 and Cys3) to validate the proteomics data, all of which showed good agreement in terms of both overall ribosome association (Figure 2A) and polysome association profiles (Figure 3C).

Reviewer #2 (Recommendations for the authors):1. Define more fully in the legend the y-axis of graph (right side) of supplemental figure 1A – % in fraction.

The Figure 1 legend has been updated and now reads:

“(A) … Right: entire lanes were quantified using LI-COR Image Studio and the percentage of the total signal that each fraction accounts for was calculated.”

2. page 7- Line 191: 465 proteins were detected in all of F1-F5, and 516 in all of F1-F4. Does this mean this number of proteins were identified to be in common in each of fractions 1 -5 or between 1-4? (ie this is not an aggregate number for this combination of fractions?).

Yes, that interpretation is correct. Fewer proteins were identified in F5, therefore more proteins were in common to all of F1-F4 than to all of F1-F5. The 516 common across F1-F4 include the 463 common across F1-F5. The text on page 7 has been updated to add more clarity:

“In unstressed conditions, 463 proteins were detected reproducibly across every fraction F1-F5, which rose to 516 when F5 was excluded (i.e. the same 516 proteins were identified in each fraction F1-F4; Figure 1E-F). Fewer proteins were found in heavy polysomes during stress when there is ribosome run off. Nevertheless, 353 proteins were identified in common in every fraction F1-F4 in all three conditions. (Figure 1E-F).”

Line 209 polysome fractions appear to be referring to F1-5, which includes monosomes, di, tri-polysomes up to heavy polysomes (F5). Totals are presumed to be materials loaded onto the gradient, but this is not clear and not detailed in the methods.

That is correct – the Totals are the cytoplasmic extracts prepared for polysome profiling, representing the total material analysed by polysome profiling. The Methods have been updated to describe this in more detail:

“Label-free MS

Cytoplasmic extracts (referred to as Totals/T) were prepared as in ‘Cell extract preparation’. These were loaded on to sucrose gradients, fractions collected, and protein extracted as in ‘Polysome profiling’ and ‘Protein extraction’. Totals were prepared for MS by adding equal volumes of sample and protein loading buffer (2x NuPAGE LDS sample buffer (Invitrogen), 715 mM 2-mercaptoethanol), and incubating for 5 min at 95°C. Both Totals and Fractions were briefly run on NuPAGE 4-12% Bis-Tris gels (Invitrogen) and protein samples were excised from the gels. Samples were dehydrated using acetonitrile and centrifuged under vacuum. …”

3. Figure 2: The figure 1A and B y-axes emphasizes % in F1-F5, while the legend indicates in F1-F4. Is F1 a polysomal fraction (monosome in Figure 1)? Operational definitions for M and P need to be clear here and in later figures.

For the analysis of the MS data, we considered the set of proteins with the broadest interest to be the 353 proteins that were common to F1-F4 in all three conditions (shown in Figure 1F). These proteins were all additionally detected in Unstressed F5, though not H2O2 F5 and/or 3-AT F5. As the data shown in Figure 2A-B are the summed fraction intensities in the Unstressed condition, for which all of this protein set had F5 data, all fractions (F1-F5) were included in the calculation for completeness.

The text, figures and legends have been updated to properly distinguish between monosomal and polysomal fractions. Fractions F1-F5 are now collectively referred to as ‘ribosomal fractions’ (previously they were referred to as ‘polysomal fractions’ in some cases), while ‘monosomes/monosomal fraction’ now refers only to F1, and ‘polysomes/polysomal fractions’ only to F2-F5. Similarly, the fraction sum-to-total ratio is now referred to as ‘estimated % ribosome association/estimated % ribosomal’ rather than ‘% polysomal’.

4. In the MS processing methods, it should be clear how statistical significance for changes in protein fraction distributions were determined.

This point is addressed in Essential revisions point 3. We have made several changes to the Methods regarding the MS data processing. Notably, we added extra processing for the data shown in Figure 2 in order to add statistical significance for changes in the ribosome association of proteins (ΔFS/T).These changes are detailed in our response to Essential revisions point 3 and in the Methods.

In addition, we investigated methods for assessing the statistical significance of clusters generated by hierarchical clustering, but this was not compatible with the original clustering we performed, as described in our response to Essential Revisions point 3. Regardless, it confirms the major trends and points to clusters with equivalent general contents and statistically significant differences to other clusters.

Reviewer #3 (Recommendations for the authors):It is a very interesting, topical, and well-executed work that reads very well and, in my opinion, meets the publishing standards of the eLife journal. It brings yet another example of until recently totally unimaginable players in translational control; i.e. metabolic enzymes. The manuscript will be of interest for all enthusiasts in the regulation of gene expression field. Below I list a few issues that might deserve the author's attention.1) Can the authors fully rule out the cross-contamination of their polysomal fractions with various stress-induced large cytoplasmic assemblies / bodies? A definitive proof of principle experiment could be to tag one of the ribosomal proteins, IP the HCHO x-linked complexes containing it, reanalyze the purified samples with MS and compare the output with no-IP data. Performing this kind of an experiment, which is easy to do in yeast, perhaps only with the most critical part of the presented data, might dismiss all doubts and, perhaps, help to explain some inconsistencies in their PE proteomics data.

We have addressed this point above in the Essential revisions point 1.

2) Figure 7. To further strengthen the argument that non-catalytic AAT2 mutants did not show heightened stress sensitivity, the authors should demonstrate that eIF2-α is not phosphorylated in these mutants (as in Figure 5A for the aat2delta strain) and measure the GCN4-lacZ (p180) activity too (as in Figure 5B).

This point is addressed in our response to Essential revisions point 2.